

# A Sedimentary Carbon Inventory for a Scottish Sea Loch (Fjord): An Integrated Geochemical and Geophysical Approach.

C. Smeaton[1], W. E. N. Austin[1,2], A. L. Davies[1], A. Baltzer[3], R. E. Abell[2] and J. A. Howe[2].

[1]{School of Geography & Geosciences, University of St-Andrews, St-Andrews, KY16 9AL, UK}

[2]{Scottish Association for Marine Science, Scottish Marine Institute, Oban PA37 1QA, UK}

[3]{Institut de Géographie et d'Aménagement Régional de l'Université de Nantes, BP 81 227 44312 Nantes cedex 3}

Correspondence to: C. Smeaton (cs244@st-andrews.ac.uk)

**Abstract**

Quantifying sedimentary carbon stocks in the coastal ocean is key to improving our understanding of long-term storage of carbon in the coastal ocean and to further constraining the global carbon cycle. Here we present a methodological approach which combines seismic geophysics and geochemical measurements to quantitatively estimate the total stock of carbon held within marine sediment. Through the application of this methodology to Loch Sunart a sea loch (fjord) on the west coast of Scotland we have created the first sedimentary carbon inventory for a fjordic system. The sediment of Loch Sunart holds $26.88 \pm 0.52$ Mt of carbon split between $11.05 \pm 0.23$ Mt and $15.02 \pm 0.35$ Mt of organic and inorganic carbon respectively. This quantative estimate of carbon stored in Loch Sunart in significantly higher than previous estimates.   Through comparison to Scottish peatland carbon stocks we have determined that Loch Sunart on a per are basis is a significantly more effective store of carbon. This initial work supports the concept that fjords are important environments for the burial and long-term storage of carbon and therefore should be considered as unique environments while considering coastal carbon stocks.



## 1 Introduction

The rising prominence of Blue Carbon, i.e. carbon (C) which is stored in coastal ecosystems, notably, mangroves, tidal marshes, seagrass meadows and sediments has forced a reassessment of our knowledge of C in the coastal ocean (Nellemann et al., 2009). In recent years there have been a number of reviews (Bauer et al., 2013 & Cai et al., 2011) highlighting knowledge gaps and the limited understanding of both the C sources and sinks in the coastal ocean (Bauer et al. 2013). Quantifying the stores of C in the coastal ocean is the first step to a better understanding of coastal carbon dynamics. Global C burial in the coastal zone is estimated in the region of 237.6 Tg yr$^{-1}$ with approximately 126.2 Tg yr$^{-1}$ of C being buried in depositional areas i.e. estuaries and the shelf (Duarte et al., 2005). The lack of regional and national coastal sedimentary C inventories means these global estimates cannot be confirmed or further constrained.

One of the rare examples of a national marine C inventory was carried out by Burrows et al. (2014) producing initial estimates of Blue Carbon in Scottish territorial waters; they calculated that these waters stored 1,757 Mt C, with coastal and offshore sediments acting as the main repositories. Burrows et al. (2014) suggested that the majority of this organic carbon (OC) was held in sea loch (fjord) sediments.

It has been long known that fjords are important stores of C (Syvitski et al., 1987) and that C burial in sediments is the most significant mechanism of long-term (>1000years) OC sequestration in the coastal ocean setting (Hedges et al., 1995).. The work by Smith et al. (2015) has also shown that globally fjordic systems act as a $CO_2$ "buffer" by efficiently capturing and burying labile terrestrially derived OC and preventing it from entering the adjacent ocean system where it is prone to recycling. These authors have calculated that 11% of annual global marine carbon sequestration occurs within fjords.

Despite these findings, much of the global research to assess and quantify C stocks is disproportionately skewed towards the terrestrial environment (e.g. Yu et al., 2010). This trend is also found at the regional scale where there have been multiple studies quantifying the carbon held within Scottish soils (Aitkenhead et al., 2016, Bradley et al., 2005, & Chapman et al., 2013) and peats (Aitkenhead et al., 2016, Howard et al., 1995, Cannell et al., 1999 & Chapman et al. 2009).





In addition to the challenges of access and cost to sample these environments when compared
to the adjacent terrestrial environment, it might also be argued that the sparsity of marine
sedimentary C inventories is due to the lack of a robust methodology to quantify these C
stores. Syvitski et al. (1987) commented that "the development of a methodological approach
to quantify the C in the sediment of a fjord must be a priority", yet in the subsequent years
there has been relatively little progress towards this goal.
The absence of a robust methodology to quantify the C held in marine sediments is illustrated
by Burrows et al. (2014), who estimated that there is 0.34 Mt OC stored in the sediments of
Scottish sea lochs (fjords). However, these calculations only take into account an estimate of
OC in the top 10 cm of sediment, despite the fact that sediment depths of >25 m are common
in Scottish sea lochs (Baltzer et al., 2010, Howe et al. 2002). Therefore, it is likely that current
best estimates of the quantity of OC has been significantly underestimated and that the
presence of significant quantities of inorganic carbon (IC) held within sea lochs sediments has
been overlooked.
This study combines geochemical, geophysical and geochronological techniques to produce a
methodology capable of delivering quantitative first-order calculations of the mass of C stored
within the sediment of a sea loch and, potentially, of achieving the goal set out by Syvitski et
al. (1987). This work provides the first carbon inventory for a fjord and further develops the
concept of these fjords as being globally important sites for the burial of C as set out by Smith
et al. (2015).
**2   Material & Methods**
**2.1   Study Area**
Loch Sunart is a sea loch on the West coast of Scotland (Fig.1). The loch is 30.7 km long and
covers an area of 47.3 km$^2$ with a maximum depth of 145 m. It consists of three basins
separated by shallower, rock sills. The inner basin is separated from the middle basin by a sill
at approximately 6 m depth, while the middle and outer basins are separated by a sill at
approximately 31 m depth (Edwards & Sharples, 1986; Gillibrand et al., 2005). The silled
nature of the bathymetry allows the loch to act as a natural sediment trap for both terrestrial
and marine derived materials (e.g. Nørgaard-Pedersen et al., 2006).
Loch Sunart's catchment covers 299 km$^2$; the main tributaries of the loch are the Rivers
Carnoch and Strontain; the latter has a mean daily discharge of 1409 m$^3$ (2009-2013). The




mean annual precipitation in Loch Sunart's catchment is 2632 ± 262 mm (Capell et al., 2013).
The combination of small catchment size and high precipitation means that the flow network
is sensitive to precipitation changes which can result in a flashy flow regime (Gillibrand et al.,

4    2005).

The catchment is largely dominated by high relief and poorly developed soils. The bedrock
consists primarily of igneous and metamorphic rocks, overlain by gley and podzol soils with
limited peat in the upper catchment (Soil Survey of Scotland., 1981). Exposed rock is
common on the steep slopes; much of the catchment's vegetation can be found by streams or
on the loch shore and is dominated by both commercial forestry and natural woodlands; there
is only very limited agriculture within the catchment. The combination of steep, exposed
slopes, poorly developed soil, a reactive river network and poorly developed vegetation
typically results in high surface runoff and sediment transport (Hilton et al., 2011).
The characteristics of Loch Sunart and its catchment are representative of sea lochs across
mainland Scotland (Edwards & Sharples., 1986), with the possible exception of Loch Etive
which has a hypoxic upper basin (Friedrich et al., 2014). The sea lochs of the Scottish Islands
(Shetland, Orkney & the Western Isles) differ from their mainland counterparts in that they
are generally shallower and have catchments characterised by lower relief and are largely
dominated by peat or peaty soil (Soil Survey of Scotland., 1981). Syvitski and Shaw's (1995)
table of generalised fjord characteristics allows us to compare the sea lochs of mainland
Scotland to other fjordic systems globally. The fjords of the Norwegian mainland, Canada and
the Fiordland, New Zealand (Hinojosa et al., 2014) are characterised by similar climate,
geomorphology, river discharge, basin water temperature and sedimentation rate as the sea
lochs of Scotland. The fjords of mainland Scotland differ significantly from those in
Greenland, Alaska, Svalbard and the Canadian Arctic, many of which still have active
glaciers, resulting in very different sediment input regime.

## 2.2   Seismic Data Acquisition and Processing

### 2.2.1   Data Acquisition

A seismic geophysical survey of Loch Sunart took place in 2002 aboard the RV *Envoy*
(Fig.2). A Seistec Boomer System was used to create seismic profile data throughout the loch.
The data were recorded using an Elics-Delph data acquisition system coupled to the
Differential Global Positioning System (DGPS). The Boomer system operated on a frequency





of 1 to 10 kHz and had a pulse duration of 75 to 250 ms at a power of 150 J. The system has a
depth resolution of 25 cm and can penetrate 100 m in soft sediment (Simpkin & Davies
1983). A total of 34 transects of the loch were acquired (Fig.2). The survey achieved an
average penetration of 50 m; gas blanking prevented the signal from penetrating the sediment
in some areas (Baltzer et al., 2010).

### 2.2.2 Defining Sedimentary Horizons

Each seismic profile was combined with the DGPS data and processed with the Petrel
(Schlumberger) software package. Subsequent analysis was undertaken using the open source
SeiSee (DMNG) software package. Initial interpolation, following Baltzer et al.'s (2010)
methodology, defined the different seismic horizons (H) and the layers between the horizons
which are defined as seismic units (U) numbered 1 to 3 from the basement horizon upwards
(Fig 3). The compilation of the horizons and units allows the construction of an equivalent
seismic stratigraphy for each sediment core and the loch as a whole.
Using SeiSee, points were picked along each of the four horizons creating polylines. Each
polyline was split into points at 0.25 m intervals and each point was assigned an x,y,z
coordinate that represents its geographic location and depth (relative to mean sea level).

### 2.3 Sediment Sampling

Eight sediment cores (Table.1) where collected from Loch Sunart (Fig.1) in 2001 using a
gravity corer (GC) as part of the HOLSMEER project. This was supplemented with further
sampling on a follow-up cruise on-board the *RV Calanus* in August 2013 where a short GC
was collected to fill a gap between the original coring sites. These cores capture the post-
glacial history of sediment accumulation within the loch, as confirmed by [14]C basal dates.
Additionally, we accessed the lower sections of core MD04 2833 which was recovered using
the CALYPSO giant piston corer from the *RV Marion Dufresne* in July 2004 as part of the
IMAGES project. Sampling of Section VIII (1050-1200 cm) of MD04 2833 was undertaken
to obtain sediment of inferred glacial origin for geochemical analysis (Baltzer et al., 2010).





**2.4 Sediment Analysis**
2.4.1 Physical Characteristics
Detailed sediment logging was undertaken for each of the cores (Supplementary Material).
The gravity cores were sub-sampled at 10 cm intervals and high resolution sampling at 1 cm
intervals was undertaken on the short core. Section VIII of glacial sediment core MD04-2833
was sub-sampled at 12 cm intervals. Each sub-sample was split for physical property and
geochemical analyses. The wet (WBD) and dry bulk density (DBD) of the sediment was
calculated following Dadey et al. (1992).
2.4.2 Bulk Elemental Analysis
To quantify the total carbon (TC) content, each sub-sample was freeze-dried and milled to a
fine powder. A $20 \pm 2$ mg aliquot was placed in a tin capsule and measured on a COSTECH
Elemental Analyser (EA) calibrated with acetanilide (Verardo et al,1990, Nieuwenhuize et al.,
1994). Precision of the analysis is estimated from repeat analysis of standard reference
material B2178 (Medium Organic content standard, Elemental Micro analysis, UK) C =
0.07% N = 0.02% (n = 8).
To quantify OC, the process was repeated with the addition of $H_2SO_3$ to remove the inorganic
carbon (IC). After acidification vessels were placed in a vacuum desiccator to remove any
remaining $CO_2$ and the sample was then freeze-dried to remove the $H_2SO_3$. IC was calculated
from the difference between TC and OC measurements. The mean standard deviation of TC
and OC triplicate measurements (n=10) were 0.04 %, 0.17 % respectively.
2.4.3 Sediment Geochronology
Basal radiocarbon dates for five of the gravity cores were obtained by accelerator mass
spectrometer (AMS) radiocarbon dating of marine carbonate material (mollusc). This was
carried out at the University of Aarhus, Denmark (AAR), Centre of Accelerator Mass
Spectrometry, USA (CAMS) and the NERC Radiocarbon Laboratory, Scotland (SUERC).
The radiocarbon dating was used to validate the Holocene chronology of the seismic
stratigraphy. A single MD04-2833 sample was processed at Laval University, Canada (UL) to
confirm that the sediment was early post-glacial in age. Dates were calibrated using OxCal
4.2.4 age modelling software (Bronk Ramsey., 2009 & Bronk Ramsey & Lee., 2013)



applying the Marine13 curve (Reimer et al., 2013) and the regional marine radiocarbon
reservoir age correction: ΔR value of -26 ± 14 yr (Cage et al. 2006).
Sediment accumulation rates (SAR) were calculated as an approximation for the whole core
using basal ages and a linear interpolation to the core top, assuming a contemporary surface.
We recognise that the calculations will be crude and do not take into consideration factors
such as compaction and possible changes in sedimentation rate, but these calculations provide
initial insight into the variability of SARs within the loch and allow first-order C
accumulation rates to be estimated.
**2.5    Sediment Quantification & Characterisation**
2.5.1  Digital Terrain Models (DTM)
The points collected from each seismic horizon were connected to form a DTM of that
horizon. This was achieved using spatial modelling techniques in ArcGIS. The compiled *x,y,z*
data were statistically tested to determine the gridding technique best suited to the
interpolation of the data. Eleven gridding techniques where subjected to cross validation
(Chiles & Delfiner 1999)( Supplementary Material).. The residual Z mean value and standard
deviation were examined; the technique with the lowest residual Z mean and standard
deviation for each horizon (and the data set as a whole) was chosen as the gridding technique
best suited to the interpolation of the data. Kriging (with linear interpolation) (Cressie, 1990)
with a 100 by 1000 node structure performed best and was chosen to create computationally
efficient DTMs for each seismic horizon.
2.5.2  Volumetric Calculations
The horizon DTM grids were used to calculate the volume of sediment in each seismic unit
and, by extension, within the loch as a whole. By subtracting one DTM grid from another
(e.g. Surface DTM – Bedrock DTM) the volume between the grids was calculated. Three
different numerical integration algorithms were used for this calculation (Eq.1,2,3). The net
volume is reported as the mean of these three calculations. In the following formulae $\Delta x$
represents the grid column spacing, $\Delta y$ represents the grid row spacing and $G_{i,j}$ represents the
grid node value in row $i$ and column $j$ (Press et al., 1988).
*Trapezoidal Rule*



The pattern of coefficients is {1,2,2,2,…,2,2,1}:                                      (1)
$A_i = \frac{\Delta x}{2}[G_{i,1} + 2G_{i,2} + 2G_{i,3} \cdots + 2G_{i,nCol-1} + G_{i,nCol}]$

3        $Volume \approx \frac{\Delta y}{2}[A_1 + 2A_2 + 2A_3 + \cdots 2A_{nCol-1} + A_{nCol}]$

*Extended Simpson's Rule*
The pattern of coefficients is {1,4,2,4,2,4,2,…,4,2,1}:                           (2)
$A_i = \frac{\Delta x}{3}[G_{i,1} + 4G_{i,2} + 2G_{i,3} + 4G_{i,4} + \cdots + 2G_{i,nCol-1} + G_{i,nCol}]$

7        $Volume \approx \frac{\Delta y}{3}[A_1 + 4A_2 + 2A_3 + 4A_3 + \cdots + 2A_{nCol-1} + A_{nCol}]$

*Extended Simpson's 3/8 Rule*
The pattern of coefficients is {1,3,3,2,3,3,2,…,3,3,2,1}:                       (3)
$A_i = \frac{3\Delta x}{8}[G_{i,1} + 3G_{i,2} + 3G_{i,3} + 2G_{1,4} + \cdots + 2G_{i,NCol-1} + G_{i,nCol}]$

11       $Volume \approx \frac{3\Delta y}{8}[A_1 + 3A_2 + 3A_3 + 2A_3 + \cdots + 2A_{nCol-1} + A_{nCol}]$

### 2.5.3 Sediment Mass Quantification

The mean dry bulk density (DBD) for each seismic unit was calculated and assigned to the
equivalent seismic units within each core. The spatial distribution of the DBD for each
seismic unit was modelled, again using Kriging (with linear interpolation). The resulting
contour plot was integrated with the volumetric model for each seismic unit to calculate the
dry mass of the sediment held within that seismic unit. The integration process calculates the
volume of sediment held within each of the DBD contours and multiplies that volume with
the associated DBD value to calculate the mass of sediment.

### 2.5.4 Sedimentary Carbon Quantification

The same methodology used to integrate the volume and density data was used to combine
bulk elemental data with the sediment dry mass calculations. Mean values for TC, OC and IC
in each seismic unit were assigned to the seismic units from the available core data. Kriging
(with linear interpolation) was again used to create contour maps representing the quantity of
TC, OC and IC in each seismic unit and the mass of sediment held between the contours was
multiplied by the percentage of the element quantifying the mass of TC, OC and IC held





within the loch's sediment. Finally, we calculated how effectively the loch stores C ($C_{eff}$) as a
depth-integrated average value per km$^2$ for both the post-glacial and glacial derived
sediments. This measure allows the loch's C stores to be directly compared with other C
stores (peatlands, soil, etc.).
**3   Results**
**3.1   Seismic Interpretation**
3.1.1 Seismic Horizons and Units
Four horizons were identified throughout the loch (Fig.3): these represent the basement (H1)
and the sediment water interface (H4) with two intermediate horizons (H2 & H3). Core
stratigraphy (Baltzer et al. 2010) indicates that H2 divides the post-glacial and glacial
sediment; while H3 splits the post-glacial sediment into two units. The seismic data displays a
fifth horizon between H1 and H2 which is only present in the inner basin and partially in the
middle basin. We interpret this as glacial sediment from the Younger Dryas, as confirmed by
radiocarbon dating (Baltzer et al., 2010, Mokeddem et al. 2010); for the purposes of this
paper, the horizon was amalgamated with H2.
A seismic stratigraphy was developed based on these horizons (Fig.3). U1 is interpreted as
glacial sediment based on the observation of the short, discontinuous seismic reflections
which are synonymous with poorly sorted material; the unit varies in thickness but never
drops below a minimum thickness of 10 m. U2 is found throughout the loch with an average
thickness of 5 – 10 m; the unit drapes over U1. U3 is the uppermost unit and has a
homogenous thickness of around 1m; it is characterised by laminated acoustic reflections.
Both U2 and U3 are interpreted as post-glacial infill of the loch; though clear in the seismic
geophysics the boundary between U2 and U3 is poorly defined in the sediment lithology
(Supplementary Material). Similar patterns in seismic stratigraphy have been observed
throughout the west coast of Scotland (Binns et al., 1974a, b, Boulton et al., 1981 and Howe
et al., 2002).
We compared our interpretation of the seismic data to the seismic interpretation of Baltzer et
al., (2010); this exercise was designed to test the replicability of our interpretation and allow
potential uncertainties in the seismic interpolation to be built into our future applications. The



comparison identified small differences in the depth of H1 (-0.17 m), H2 (+0.34) & H3 (-0.22
m). These differences were integrated into the volumetric calculations as an error term.

## 3.2 Sediment Geochronology

Calibrated radiocarbon dates for the gravity cores (Table.2) indicate that these cores are
comprised of sediment accumulated during the post-glacial period (Holocene). The age of the
deeper basal sediment of MD04-2833(Section VIII) was confirmed through dating of a
mollusc (*Pecten maximus*); the calibrated age was 17041 ± 312 cal BP which, combined with
the characteristic glacial core lithology of poorly sorted sedimentary material, indicates that
this basal sediment of MD04-2833 was deposited by the retreat of the British ice sheet (BIS)
at the end of the last glacial period 13500 to 17000 cal BP (Clark et al., 2010, Scourse et al.,
2009, Wilson et al. 2002).
Through comparison of the chronologies to the seismic stratigraphy we can test the
interpolation and further constrain the age of each seismic unit. The seismic unit for the
equivalent depth of each of the radiocarbon samples has been compiled (Table.2), then
compared to the seismic unit that the sample would fall into based on age alone as per the
Baltzer et al. (2010) chronostragraphy. Of the 18 samples tested, 15 have a matched pair of
seismic units; the three that do not have corresponding seismic units are all from GC023,
suggesting a problem with the dating of this core rather than the interpolation of the seismic
geophysics. This test signifies that our interpolation of the seismic geophysics is accurate and
that the chronostratigraphy developed for MD04-2833 (Baltzer et al., 2010) can be applied
throughout Loch Sunart. The seismic interpolation and the dated samples confirm that both
U2 and U3 are postglacial in origin. We can further constrain the age of the seismic units with
U2 representing the early to mid-Holocene and U3 mid to late Holocene in age.
SARs vary between the sedimentary basins, with the most rapid rates in the middle basin as
indicated by GC023 ($0.12 \pm 0.026$ cm yr$^{-1}$). The inner basin has a similar SAR ($0.089 \pm 0.026$
cm yr$^{-1}$) to the middle basin, but accumulation rates were significantly slower in the outer
basin, as illustrated by GC011 ($0.017 \pm 0.006$ cm yr$^{-1}$).



### 3.3 Sediment Analysis

### 3.3.1 Bulk Density Measurement

Mean DBD was calculated for U1, U2 and U3 from each core. Figure 4 displays the DBD results, which are arranged to mirror the spatial distribution of the cores, from the inner basin to the outer basin. U1 sediment is characterised by the single section of MD04-2833, which has a mean DBD of $2.19 \pm 0.09$ g cm$^{-3}$. This is within the range of other northern hemisphere fjords (Pedersen et al. 2012, Forwick et al. 2010 and Baeten et al. 2010). DBD increases down each core as a result of sediment dewatering in response to compaction. GC011 is the only core where U3 has a higher DBD than U2 most likely due to large quantities of shell in the upper part of the core. U1 has the highest DBD; this reflects both the type of sediment deposited during glacial retreat and long-term compaction over the post-glacial period.

### 3.3.2 Bulk Elemental Analysis

The mean quantity OC and IC has been calculated for U1, U2 and U3 (Fig.5). Again values for U1 have been calculated using basal sediments of MD04-2833 (Section VIII). Clear trends emerge from the data, with U3 always containing a greater quantity of OC than U2, while the proportion of sedimentary OC generally decreases away from the inner basin. The opposite is true for sedimentary IC, which generally increases away from the inner basin. Sediment Quantification & Characterisation

### 3.3.3 Digital Terrain Models (DTMs)

The interpolation of the seismic profiles led to the creation of four DTMs (Fig.6) which represent horizons H1 to H4. Both H4 and H3 follow the trajectory with minimal deviation, primarily because U3 has a uniform depth of 1 m throughout the loch. Differences in depth between H3, H2 and H1 are far more variable. The inner basin shows the least change in depth between the horizons, although both H2 and H1 deepen before the sill (Fig.5). The middle basin displays the greatest depth differences between the horizons and also where the majority of the sediment is stored. Patterns in the outer basin are similar to those in the middle basin, especially where the two meet. Horizon depths become less variable towards the seaward direction.

To determine the accuracy of the models, the DTM for H4 was compared to an existing high-resolution bathymetric model of the loch (Bates et al. 2004). The coordinates ($x,y,z$) of key





high and low points (n=12) were compared between surveys; the mean divergence between
surveys were calculated as  $x$: -0.56 m , $y$: -0.81 , $z$: 0.21. Although the H4 DTM slightly
negatively offsets the $x,y$ and overestimates the $z$ coordinates of these points, the general
location and pattern of these seabed features compare favourably.

### 3.3.4  Volumetric Modelling

The DTMs and numerical integration algorithms were combined to calculate the volume of
sediment held within each seismic unit. This was further broken down by basin and into post-
glacial (U2 & U3) and glacial (U1) derived sediment (Table.3). The loch as a whole contains
a greater volume of glacial (599731882 m$^3$ ± 1.89 %) than post-glacial sediment (530872293
m$^3$ ± 7.39 %). The outer basin is the only area where this trend is reversed. Comparison of the
three basins indicates that the middle basin contains the greatest combined (post-glacial +
glacial) volume of sediment (30409301.04 m$^3$ ± 5.30 %) followed by the outer (16039257.2
m$^3$ ± 5.74 %) and inner basins (4171662.46 m$^3$ ± 4.48 %).

### 3.3.5  Sediment Mass Quantification

The mean DBD for U2 and U3 were modelled (Fig.7) to determine the variability in spatial
distribution throughout the loch. A similar spatial pattern of DBD is found in both U2 and U3;
the DBD is lowest in the inner basin (U2: 0.47 g cm$^{-3}$, U3: 0.59 g cm$^{-3}$) rising out through the
middle basin where it peaks at 1.75 g cm$^{-3}$ and 1.67 g cm$^{-3}$ for U2 and U3 respectively.  The
transition between the middle and outer basins is characterised with low DBD values (U2:
0.72 g cm$^{-3}$, U3: 0.91 g cm$^{-3}$); from this low point the DBD rises towards the seaward end of
the loch.
The model output was integrated with the volumetric data to calculate the mass of sediment
held within post-glacial sediment (Table 4). Since we have a single mean value for DBD for
U1 we applied this throughout the loch to calculate the mass of sediment held within this unit.
The loch holds a total of 1928.26 ± 7.29 Mt of sediment which is split into 652.09 ± 6.62 Mt
of post-glacial and 1276.17 ± 8.93 Mt of glacial sediment. The inner basin holds the least
sediment followed by the outer basin with the middle basin acting as the main store of
sediment in Loch Sunart.





### 3.3.6 Sedimentary Carbon Quantification

Using a similar approach, the mean OC and IC were spatially modelled throughout the loch. The output for U3 is illustrated in Figure 8. As before, the model outputs for U2 and U3 were integrated with the sediment mass data in order to quantify the mass of TC, OC and IC held within the post-glacial and glacial sediment (Table.4). Single mean values for TC, OC and IC were again used to calculate their respective mass of C within the sediment of U1.

The sediment of Loch Sunart holds a significant quantity of C ($26.88 \pm 0.52$ Mt) split between OC ($11.05 \pm 0.23$ Mt) and IC ($15.02 \pm 0.35$ Mt). Though a greater mass of sediment is held within the glacial sediment component, it is the post-glacial sediments which hold the largest quantity of C ($19.88 \pm 0.27$ Mt). The quantity of C held within each of Loch Sunart's basins varies; the lowest amount is found in the inner basin ($2.12 \pm 0.45$ Mt) followed by the outer basin ($6.70 \pm 0.64$ Mt). The sediment of middle basin holds significantly more C than both the inner and outer basins combined; with $18.05 \pm 0.66$ Mt C stored in these sediments indicating that the middle basin is the main repository for sedimentary C in Loch Sunart.

How effectively the loch stores C is measured by the $C_{eff}$ (Table.5) and the OC:IC ratio. Loch Sunart is characterised by an OC:IC ratio of 0.74 and has a $C_{eff}$ of 0.568 Mt C km$^{-2}$, which can be further broken down to a post-glacial $C_{eff}$ of 0.42 Mt C km$^{-2}$ and a glacial $C_{eff}$ of 0.148 Mt C km$^{-2}$. The effective C storage can also be illustrated at the individual basin level with the post-glacial sediments of the inner, middle and outer basins characterised by OC:IC ratios of 0.42, 1.00 and -0.42, illustrating the transition from OC as the dominant component of the sediment in the upper loch to an IC-dominated sediment at the seaward end of the loch. The middle basin is the most effective at storing post-glacial OC followed by the inner and outer basin; similarly the middle basin is most effective at storing IC, but in contrast to the effective storage of OC, the outer basin ranks second followed by the inner basin for IC. The glacial material held within the loch as a whole is characterised by an OC:IC ratio of 0.42 with a mean $OC_{eff}$ 0.044 Mt km$^{-2}$ and $IC_{eff}$ 0.104 Mt km$^{-2}$.

## 3.4 A Methodology for Estimating Sedimentary Carbon and Attributing Uncertainty Estimates

The joint geophysical and geochemical methodology outlined (Fig.9) provides a robust approach to allow the first quantification of sedimentary C stocks in a fjord setting. An important part of estimating sedimentary C stocks should be the quantification of uncertainty



associated with these estimates. There are several types of uncertainty that can influence
sedimentary carbon estimations (Fig.9), including interpolation, algorithmic, analytical,
sampling and extrapolation uncertainty. Several of these types of uncertainties are easily dealt
with statistically, for example the analytical uncertainties can be quantified through triplicate
measurements. The sampling uncertainty of a stratigraphic sequence (i.e. spatial variability of
C content in relation to sampling density) can be overcome by calculating the mean and
standard deviation to create composite values that are representative of the entire sediment or
seismic unit. We integrated the quantifiable uncertainties at each calculation step (Fig.4). By
calculating composite standard deviations we are able to propagate the uncertainties
throughout the C quantification process. In the interpolation of the seismic geophysics, it is
difficult to fully quantify the uncertainty involved in the process.  Bond et al. (2007) set out a
5 step framework designed to reduce uncertainty in this process. We utilised the framework of
Bond et al. (2007) and additionally integrated a validation step using radiocarbon dating of
sedimentary cores (See Section 3.2). This allows us to reduce the uncertainties associated
with the seismic interpretation, although we recognise that some uncertainty remains (e.g.
highly variable patterns of depth) which cannot be fully quantified.  Within this framework of
uncertainty, we consider our method to give a robust estimate for the carbon socks present.

## 4    Discussion: A new Sedimentary C Inventory for Scottish Coastal Waters

The development of this methodology has allowed the estimation of the sedimentary C stocks
stored in Loch Sunart. The sediment which has been uncounted for within Loch Sunart holds
26.88 Mt C.
The only directly comparable estimation for sedimentary C stocks is the report by Burrows et
al. (2014), where they calculated that 0.34 Mt OC was stored in all Scottish sea lochs. In
comparison, our findings estimate that Loch Sunart alone holds 11.05 Mt OC. However,
Burrows et al. (2014) focused on the top 10 cm of sediment because data availability and the
lack of a robust methodology made it impossible to calculate the entire sedimentary C stock;
this has resulted in a significant underestimation of the quantity of C held within the sediment
of these lochs.  Additionally, Borrows et al. (2014) did not consider IC to be a major
component in these sediments; instead the authors focused on Scottish sea lochs largely as OC
stores.  In contrast, our results demonstrate that Loch Sunart stores 15.02 Mt IC in comparison
to 11.05 Mt OC. The general lack IC data for Scottish sea lochs makes it difficult to assess
how representative Loch Sunart is of these coastal sedimentary IC stores; our results do



highlight the potential significance of IC as a major component of sedimentary C stores in
these depositional environments. Our results also highlight that sea lochs (and probably fjords
in general) act as an OC-rich sediment transition zone between terrestrial and oceanic
environments.
Loch Sunart's sediment currently holds 11.05 Mt OC which has been trapped and prevented
from reaching the adjacent shelf sea. This OC trapping in the coastal zone may reduce
reworking and remineralisation of the material which would have otherwise resulted in the
release of $CO_2$ through biotic processes (Smith et al., 2015). This 11.05 Mt OC is equivalent
to 40.93 Mt $CO_2$e (carbon dioxide equivalent). As a whole, the sediment within Loch Sunart
stores 99.56 Mt $CO_2$e which is almost double Scotland's total greenhouse gas emission for
2013 which reached an estimated 53 Mt $CO_2$e (Scottish Government, 2015).
Globally, the terrestrial C stores have received much more attention than their marine
counterparts; with significant focus on quantifying the forest (Köhl et al., 2015) and soil C
stocks (Köchy et al., 2015, Scharlemann et al., 2014). The work by Duarte et al. (2005) to
compile the known stocks and burial rate of C in the coastal environment highlighted that the
coastal ocean is a large store of carbon, which remains poorly understood; from this work the
concept of Blue Carbon arose (Nellemann et al., 2009). The focus of Duarte et al. (2005) was
to highlight that the vegetated coastal zones (i.e. saltmarsh, seagrass and mangroves) bury and
store significant quantities of C and that these stores should be further investigated and
recognised in policy outputs, but they largely overlooked the importance of what they
described as depositional area (estuaries and the shelf sea) as long-term repositories of OC
detritus from the vegetated coastal environment (Krumhansl et al. 2012). These authors
recognised that coastal (and shelf?) depositional areas are important stores of sedimentary C
globally, yet no consideration is given to how these areas vary in terms of their capacity to
store C. Conceptually, if we consider the types of estuaries (i.e. fjord, delta, coastal plain, bar-
built and tectonic), it is clear that the characteristics of each type of estuary will impact the
manner in which C is buried and stored, for example the restricted nature of fjords will be
conducive to sediment capture and C storage compared to the more open estuarine types.
Our initial work suggests that the depositional area category could be further expanded upon
to include fjords as a separate component and this concept is supported by Smith et al. (2015),
who indicated that fjords are "hot-spots for OC burial" and should be considered separately
from estuaries when investigating global ocean OC burial. Currently, there is insufficient data





globally to advocate fjords being categorised as a separate component in global coastal C
stores; the standardised methodology outlined (Fig.4) provides a platform to investigate this
concept further.
At the national level there has been a significant focus on quantifying Scottish soil C stocks,
with much attention given to the peatlands (Aitkenhead & Coull., 2016, Bradley et al., 2005
& Chapman et al. 2009). Peat and other organic rich soils cover 66% of Scotland and account
for 50% of all the United Kingdom's soil C stocks (Cummins et al., 2011). The Scottish
peatlands store 1620 Mt C (Chapman et al., 2009) over an area of 17270 km$^2$, while the other
soils hold 2110.9 Mt C over 60215 km$^2$ (Aitkenhead & Coull., 2016). In comparison to these
figures, the quantity of C stored in Loch Sunart is small, but the loch itself only covers an area
of 47.3 km$^2$. When the loch's $C_{eff}$ is compared to how effectively Scotland's soils and
peatland store C (Table.5) we can see that on a C amount per unit area basis Loch Sunart
stores significantly more C than the soils of Scotland. The loch has a $C_{eff}$ of 0.568 Mt C km$^{-2}$
compared to 0.094 Mt C km$^{-2}$ and 0.035 Mt C km$^{-2}$ for the peatlands and other soils of
Scotland. Our results suggest that Loch Sunart is one of the most effective stores of C in
Scotland and highlights the potential of the sediment in these sea lochs to hold a significant
quantity of C which has previously not been recognised. Many of these terrestrial C stores
are, of course, vulnerable to rapid and long-term environmental change; the Scottish terrestrial
C stocks are at risk from erosion (Cummins et al. 2011) and even fire (Davies et al. 2013),
both of which are increasing in pace and frequency by anthropogenic activities. In
comparison, a sea loch's geomorphology combined with its depth gives sedimentary C stores
a level of protection not afforded to terrestrial C stores. This does not mean that the
sedimentary C in sea lochs is invulnerable, but rather that it is buffered from the immediate
effects of chemical, biological and physical environmental change. Little is currently known
regarding the long-term stability of these stores.
The methodology outlined in this paper has given us a platform to calculate the carbon stocks
within Loch Sunart and has the potential to be applied in other fjordic systems as well as
environments with restricted sediment exchange processes, such as estuaries and freshwater
lakes, as well as artificial systems such as reservoirs and irrigation pools.
**5   Conclusion**
The integration of the geochemical and geophysical techniques outlined provides a robust and
repeatable methodology to quantitatively calculate the volume of sediment and make first





order estimations of carbon stored within fjordic sediments. Using this methodology we have
shown that Loch Sunart holds 26.88 Mt C which is almost double the quantity of Scottish
$CO_2$ emissions for 2013. Although this is small in comparison with Scotland's peatland and
soil C stocks, per unit area Loch Sunart is a more effective store of both OC and IC than
Scotland's soils or peatlands. The results from this study suggest that the sediment in
Scotland's 110 sea lochs (Edwards and Sharples et al., 1986) represent a potentially
significant, yet currently largely unaccounted for repository for both OC and IC. These
coastal settings trap and prevent the remineralisation of OC into the atmosphere. Additionally,
the C held within these 110 sea lochs is likely to represent a significant part of Scotland's blue
carbon capital that has not been considered at the marine ecosystem, global C cycle and wider
policy levels. Without a better understanding of these stores of marine sedimentary C we will
remain unable to fully quantify the coastal C cycle or the role that these fjordic environments
play in buffering the release of $CO_2$ through the burial of C in these sediments. The future
strategic use of this methodology within different fjord types and locations offers the potential
to upscale and quantify the C held within all Scottish sea lochs and possibly begin to estimate
the fjordic sedimentary C stores both at a national and global level.
**Author Contribution**
CS & WA conceived the research and wrote the initial manuscript, to which all co-authors
contributed data or provided input. CS conducted the research as part of his PhD at the
University of St Andrews, supervised by WA, AD and JH.
**Acknowledgements**
This work was supported by the Natural Environment Research Council [Grant Number:
NE/L501852/1] with additional support from the NERC Radiocarbon Facility [Allocation
1934.1015]. Seismic profiles and the CALYPSO long core were acquired within the frame of
the French ECLIPSE program. The authors would like to thank Marion Dufresne's Captain J.-
M. Lefevre, the Chief Operator Y. Balut (from IPEV) and Richard Bates (University of St
Andrews). Additionally; we would like to thank Colin Abernethy (Scottish Association of
Marine Science) for laboratory support.
Author(s) 2016. CC-BY 3.0 License.



Bronk Ramsey, C. and Lee, S. Recent and planned developments of the program OxCal.
Radiocarbon, 55(2-3), 720-730, 2013
Burrows M.T., Kamenos N.A., Hughes D.J., Stahl H., Howe J.A. and Tett P. Assessment of
carbon budgets and potential blue carbon stores in Scotland's coastal and marine
environment. Scottish Natural Heritage Commissioned Report No. 761, 2014
Cage, A.G., Heinemeier, J. and Austin, W.E.N. Marine radiocarbon reservoir ages in Scottish
coastal and fjordic waters, Radiocarbon, Vol 48, Nr 1, 31–43, 2006.
Cai, W.-J.: Estuarine and coastal ocean carbon paradox: CO2 sinks or sites of terrestrial
carbon incineration?, Ann. Rev. Mar. Sci., 3, 123–45, doi:10.1146/annurev-marine-120709-
10 142723, 2011.

Capell, R., Tetzlaff, D. and Soulsby, C.: Will catchment characteristics moderate the
projected effects of climate change on fl ow regimes in the Scottish Highlands ?, ,
699(December 2012), 687–699, doi:10.1002/hyp.9626, 2013.
Cannell, M.G.R., Milne, R., Hargreaves, K.J., Brown, T.A.W., Cruickshank, M.M., Bradley,
R.I., Spencer, T., Hope, D., Billett, M.F., Adger, W.N. and Subak, S.  National inventories of
terrestrial carbon sources and sinks: The UK experience. Climatic Change, 42, 505–530, 1999
Clark, C. D., Hughes, A. L. C., Greenwood, S. L., Jordan, C. and Petter, H.: Pattern and
timing of retreat of the last British-Irish Ice Sheet, Quat. Sci. Rev.,
doi:10.1016/j.quascirev.2010.07.019, 2010.
Chapman, S.J., Bell, J.S., Campbell, C.D., Hudson, G, Lilly, A., Nolan, A.J., Robertson,
A.H.J., Potts, J.M., and Towers, W, Comparison of soil carbon stocks in Scottish soils
between 1978 and 2009, European Journal of Soil Science, 64, (4), pp. 455–465, 2013
Chapman, S.J., Bell, J., Donnelly, D. and Lilly, A., Carbon stocks in Scottish peatlands, Soil
Use and Management, 25, (2), pp. 105–112, 2009.
Chiles, J.P., and Delfiner.P. Geostatistics: Modeling Spatial Uncertainty. John Wiley and
Sons, New York, 695, 1999.
Cressie, N.A.C.  The Origins of Kriging, Mathematical Geology, v. 22, p. 239-252, 1990.
Cummins, R., Donnelly, D., Nolan, A., Towers, W., Chapman, S., Grieve, I. and Birnie, R.V.
Peat erosion and the management of peatland habitats. Scottish Natural Heritage
Commissioned Report No. 410, 2011



Dadey, K.A., Janecek, T. and Klaus, A Dry bulk density: its use and determination,
Proceedings of the Ocean Drilling Program, Scientific Results, Vol. 126, 1992.
Davies, G. M., Gray, A., Rein, G. and Legg, C. J.: Peat consumption and carbon loss due to
smouldering wildfire in a temperate peatland, For. Ecol. Manage., 308, 169–177,
doi:10.1016/j.foreco.2013.07.051, 2013.
Duarte, C. M., Middelburg, J. J. and Caraco, N.: Major role of marine vegetation on the
oceanic carbon cycle, Biogeosciences, 2, 1–8, 2005.
Edwards,A.& Sharples, F. Scottish Sea Lochs: A Catalogue. Scottish Marine Biological
Association/ Nature Conservancy Council, Oban, 1986.
Forwick, M., Vorren, T. O., Hald, M., Korsun, S., Roh, Y., Vogt, C. and Yoo, K.-C.: Spatial
and temporal influence of glaciers and rivers on the sedimentary environment in
Sassenfjorden and Tempelfjorden, Spitsbergen, Geol. Soc. London, Spec. Publ., 344(1), 163–
193, doi:10.1144/SP344.13, 2010.
Friedrich, J., Janssen, F., Aleynik, D., Bange, H. W., Boltacheva, N., Çagatay, M. N., Dale,  a.
W., Etiope, G., Erdem, Z., Geraga, M., Gilli,  a., Gomoiu, M. T., Hall, P. O. J., Hansson, D.,
He, Y., Holtappels, M., Kirf, M. K., Kononets, M., Konovalov, S., Lichtschlag,  a.,
Livingstone, D. M., Marinaro, G., Mazlumyan, S., Naeher, S., North, R. P., Papatheodorou,
G., Pfannkuche, O., Prien, R., Rehder, G., Schubert, C. J., Soltwedel, T., Sommer, S., Stahl,
H., Stanev, E. V., Teaca,  a., Tengberg,  a., Waldmann, C., Wehrli, B. and Wenzhöfer, F.:
Investigating hypoxia in aquatic environments: Diverse approaches to addressing a complex
phenomenon, Biogeosciences, 11(4), 1215–1259, doi:10.5194/bg-11-1215-2014, 2014.
Gillibrand, P.A., Cage, A.G. and Austin, W.E.N. A preliminary investigation of basin water
response to climate forcing in a Scottish fjord: evaluating the influence of the NAO,
Continental Shelf Research, 25, (5-6), pp. 571–587, 2005
Hedges, J.I., Keil, R.G. and Benner, R. What happens to terrestrial organic matter in the
ocean? Organic Geochemistry. 27, 195–212, 1997.
Hilton, R.G., Galy, A., Hovius, N. & Horng, M.J. Efficient transport of fossil organic carbon
to the ocean by steep mountain rivers: An orogenic carbon sequestration mechanism. Geology
39, 71–74, 2011.





Hinojosa, J.L, Christopher M. Moy, C.M, Claudine H. Stirling, C.H, Gary S. Wilson, G.S,
and Eglinton, T.I : Carbon cycling and burial in New Zealand's fjords, , 4047–4063,
doi:10.1002/2014GC005433.Received, 2014.
Howard, P.J.A., Loveland, P.J., Bradley, R.I., Dry, F.T., Howard, D.M. and Howard, D.C..
The carbon content of soil and its geographical distribution in Great Britain. Soil Use and
Management, 11, 9–15, 1995.
Howe, J. A., Shimmield, T., Austin, W. E. N. and Longva, O.: Post-glacial depositional
environments in a mid-high latitude glacially-overdeepened sea loch , inner Loch Etive ,
western Scotland, , 185, 417–433, 2002.
Johnston, D.H and R. Cooper, M.R., Methods and Applications in Reservoir Geophysics,
Investigations in geophysics, no. 15., Tulsa, OK : Society of Exploration Geophysicists, 2010.
Kennedy, P., Kennedy, H., and Papadimitriou, S. The effect of acidification on the
determination of organic carbon, total nitrogen and their stable isotopic composition in algae
and marine sediment, Rapid Communications in Mass Spectrometry, 19, (8), pp. 1063–1068,
15  2005.

Köchy, M., Hiederer, R. and Freibauer, A.: Global distribution of soil organic carbon – Part 1:
Masses and frequency distributions of SOC stocks for the tropics, permafrost regions,
wetlands, and the world, Soil, 1(1), 351–365, doi:10.5194/soil-1-351-2015, 2015.
Köhl, M., Lasco, R., Cifuentes, M., Jonsson, ??rjan, Korhonen, K. T., Mundhenk, P., de Jesus
Navar, J. and Stinson, G.: Changes in forest production, biomass and carbon: Results from the
2015 UN FAO Global Forest Resource Assessment, For. Ecol. Manage., 352, 21–34,
doi:10.1016/j.foreco.2015.05.036, 2015
Krumhansl, K. A. and Scheibling, R. E.: Production and fate of kelp detritus, Mar. Ecol. Prog.
Ser., 467, 281–302, doi:10.3354/meps09940, 2012
Mokeddem, Z., Baltzer, A.., Goubert, E and Clet-Pellerin, M., A multiproxy
palaeoenvironmental reconstruction of Loch Sunart (NW Scotland) since the Last Glacial
Maximum, Geological Society, London, Special Publications, 344, (1), pp. 341–353, 2010.
Nellemann C., Corcoran E., Duarte C.M., Valdés L., DeYoung C., Fonseca L., Grimsditch G.
(Eds.), Blue Carbon: A Rapid Response Assessment, United Nations Environment
Programme, GRID-Arendal, (2009).



Nieuwenhuize, J., Maas, Y.E.M., and Middelburg, J.J. Rapid analysis of organic carbon and
nitrogen in particu- late materials. Mar. Chem. 45:217-224, 1994
Nørgaard-Pedersen, N., Austin, W. E. N., Howe, J. a. and Shimmield, T.: The Holocene
record of Loch Etive, western Scotland: Influence of catchment and relative sea level changes,
Mar. Geol., 228(1-4), 55–71, doi:10.1016/j.margeo.2006.01.001, 2006.
Pedersen, J. B. T., Kroon, a., Jakobsen, B. H., Mernild, S. H., Andersen, T. J. and Andresen,
C. S.: Fluctuations of sediment accumulation rates in front of an Arctic delta in Greenland,
The Holocene, 23(6), 860–868, doi:10.1177/0959683612474480, 2013.
Pergamon Press, 119–41Polson, D. and Curtis, A.: Dynamics of uncertainty in geological
interpretation, J. Geol. Soc. London., 167(1), 5–10, doi:10.1144/0016-76492009-055, 2010.
Press, W.H., Flannery, B.P., Teukolsky, S.A., and Vetterling, W.T. Numerical Recipes in C,
Cambridge University Press, 1988.
Reimer, P, IntCal13 and Marine13 Radiocarbon Age Calibration Curves 0–50,000 Years cal
BP, Radiocarbon, 55, (4), pp. 1869–1887, 2013.
Scottish Government. 2015. http://www.gov.scot/Publications/2015/06/1939; Accessed
16 25/11/2015.

Scharlemann, J. P., Tanner, E. V., Hiederer, R. and Kapos, V.: Global soil carbon:
understanding and managing the largest terrestrial carbon pool, Carbon Manag., 5(1), 81–91,
doi:10.4155/cmt.13.77, 2014.
Scourse, J. D., Haapaniemi, A. I., Colmenero-hidalgo, E., Peck, V. L., Hall, I. R., Austin, W.
E. N., Knutz, P. C. and Zahn, R.: Growth , dynamics and deglaciation of the last British –
Irish ice sheet : the deep-sea ice-rafted detritus record, Quat. Sci. Rev., 28(27-28), 3066–3084,
doi:10.1016/j.quascirev.2009.08.009, 2009
Soil Survey of Scotland Staff. (1970-1987). Soil maps of Scotland (partial coverage) at a scale
of 1:25 000. Macaulay Institute for Soil Research, Aberdeen.
Simpkin,P.G. and Davis, A,  For seismic profiling in very shallow water, a novel receiver. In
Sea Technology, 1983.
Smith, R.W., Bianchi, T,S., Allison, M., Savage, C. & Galy, V, High rates of organic carbon
burial in  fjord sediments globally, Nature, doi: 10.1038/NGEO2421, 2015.



Syvitski, J.P.M, Burrell, D.C & Skei, J.M. Fjords, Processes and Products, Springer-Verlag
New York, 1987.
Syvitski, J.P.M & Shaw, J: Sedimentology and Geomorphology of Fjords, Geomorphology
and Sedimentology of Estuaries. Developments in Sedimentology 53, 1995.
Wilson, L. J., Austin, W. E. N. and Jansen, E.: The last British Ice Sheet : growth , maximum
extent and deglaciation, , (2001), 243–250, 2002.
Verardo, D.J., P. N. Froelich, P.N. and McIntyre, A,Determination of organic carbon and
nitrogen in marine sediments using the Carlo Erba NA-1500 Analyzer. Deep Sea Res. 37:157-

9    165, 1990

Yu, Z, Loisel, J., Brosseau, D.P., Beilman, D.W., & Hunt, S.J. Global peatland dynamics
since the Last Glacial Maximum. Geophysical Research Letters 37, L13402, 2010.





1 **Table.1.** Details of the sediment cores extracted from Loch Sunart that were used in this study.

| Core ID | Basin | Position (Lat, Long) | Water Depth (m) | Recovery (m) |
|---|---|---|---|---|
| GC009 | Middle | 56.672056, -5.867083 | 107 | 1.41 |
| GC011 | Outer | 56.759861, -5.969639 | 91 | 2.45 |
| GC013 | Inner | 56.681306, -5.629528 | 58 | 1.67 |
| GC016 | Inner | 56.680944, -5.642333 | 58 | 0.56 |
| GC020 | Middle | 56.704278, -5.751333 | 105 | 2.38 |
| GC022 | Middle | 56.680333, -5.804944 | 120 | 2.46 |
| GC023 | Middle | 56.665917, -5.840361 | 87 | 2.89 |
| GC081 | Middle | 56.668972, -5.863278 | 58 | 3.63 |
| GC01 | Middle | 56.696806, -5.704972 | 42 | 0.21 |
| | | | | |
| MD04 2833 | Middle | 56.665500, -5.859667 | 38 | 12 |



**Table.2** Radiocarbon ages from Loch Sunart cores. Ages were calibrated using OxCal 4.2.4 (Bronk
Ramsey., 2009 & Bronk Ramsey & Lee., 2013) with the Marine13 curve (Reimer et al. 2013) and
regional correction of ΔR value of -26 ± 14 yr (Cage et al. 2006) . All ages are calibrated at 95.4%
probability and the mean age has been determined from the minimum and maximum calibrated ages.
Additionally; we list the seismic unit assigned to each equivalent (eqv.) depth of each sample and
compare this to the age equivalent seismic unit based on Baltzer et al. (2010).

| Laboratory Code | Core ID | Depth (cm) | $^{14}$C Age, BP (No Correction) | Calibrated $^{14}$C Age (cal BP) | Seismic Unit | |
|---|---|---|---|---|---|---|
| | | | | | Depth eqv. | Age eqv. |
| AA-48108 | GC009 | 140 | 9827 ± 49 | 10801 ± 93 | U2 | U2 |
| SUERC 65990 | GC011 | 60 | 2837 ± 35 | 2625 ± 66 | U3 | U3 |
| SUERC 65991 | GC011 | 120 | 9890 ± 38 | 10878 ± 87 | U2 | U3 |
| SUERC 65992 | GC011 | 170 | 11266 ± 40 | 12760 ± 61 | U2 | U2 |
| AA-48109 | GC011 | 231 | 12181 ± 58 | 13658 ± 90 | U1 | U1 |
| AA-48107 | GC013 | 113 | 1716 ± 32 | 1294 ± 35 | U3 | U3 |
| SUERC 65995 | GC016 | 30 | 1865 ± 35 | 1438 ± 51 | U3 | U3 |
| SUERC 65994 | GC020 | 9 | 683 ± 35 | 357 ± 44 | U3 | U3 |
| SUERC 65993 | GC020 | 19 | 3067 ± 37 | 2864 ± 57 | U3 | U3 |
| AA-48106 | GC020 | 126 | 11652 ± 74 | 13160 ± 90 | U2 | U2/U1 |
| AA-51569 | GC023 | 30 | 340 ± 60 | 64 ± 51 | U3 | U3 |
| SUERC-681 | GC023 | 49 | 1215 ± 47 | 788 ± 58 | U3 | U3 |
| SUERC-677 | GC023 | 58 | 1322 ± 43 | 886 ± 55 | U3 | U3 |
| AA-51570 | GC023 | 73 | 1430 ± 55 | 1011 ± 66 | U3 | U3 |
| SUERC-679 | GC023 | 111.5 | 1695 ± 57 | 1274 ± 59 | U2 | U3 |
| SUERC-680 | GC023 | 250 | 2180 ± 61 | 1801 ± 80 | U2 | U3 |
| CAMS-82821 | GC023 | 286 | 2425 ± 40 | 2099 ± 70 | U2 | U3 |
| UL 2853 | MD04-2833 | 745 | 14420 ± 210 | 17041 ± 312 | U1 | U1 |





**Table.3** Sediment volume calculated as the mean of the three numerical integration algorithms; the
error is reported as relative standard deviation (%RSD) which integrates the uncertainty in the seismic
interpolation and the standard deviation of the numerical integration algorithms. The data is reported
for the post-glacial (PG) and glacial (G) sediment at the basin level.

| Basin | Layer | Volume | |
|---|---|---|---|
| | | Mean (m$^3$) | %RSD |
| Inner | PG | 2869825.90 | 6.48 |
| | G | 1301836.56 | 1.89 |
| Middle | PG | 23046267 | 7.26 |
| | G | 7363034.04 | 1.89 |
| Outer | PG | 13371884 | 7.90 |
| | G | 2667373.2 | 1.89 |
| Loch Sunart | PG | 530872293 | 7.39 |
| | G | 599731882 | 1.89 |
| | **Total** | **1130604175.55** | **3.61** |





1  **Table.4** Mass of sediment held within Loch Sunart and the mass of total carbon (TC), organic carbon

2  (OC) and inorganic carbon (IC) held with Loch Sunart's Sediment.

| Basin | Layer | Mass (Mt) | TC (Mt) | OC (Mt) | IC (Mt) |
|---|---|---|---|---|---|
| Inner | PG | 27.05 ± 2.98 | 1.31 ± 0.16 | 1.05 ± 0.14 | 0.26 ± 0.15 |
| | G | 126.65 ± 7.15 | 0.81 ± 0.62 | 0.24 ± 0.2 | 0.57 ± 0.41 |
| Middle | PG | 421.53 ± 7.26 | 14.08 ± 0.30 | 7.05 ± 0.25 | 7.03 ± 0.17 |
| | G | 738.31 ± 9.62 | 3.97 ± 0.88 | 1.17 ± 0.28 | 2.80 ± 0.57 |
| Outer | PG | 203.52 ± 11.10 | 4.49 ± 0.34 | 1.32 ± 0.14 | 3.15 ± 0.24 |
| | G | 411.22 ± 9.78 | 2.21 ± 0.84 | 0.65 ± 0.14 | 1.56 ± 0.57 |
| Loch Sunart | PG | 652.09 ± 6.62 | 19.88 ± 0.27 | 8.98 ± 0.21 | 10.09 ± 0.17 |
| | G | 1276.17 ± 8.93 | 7.00 ± 0.82 | 2.07 ± 0.27 | 4.93 ± 0.55 |
| | **Total** | **1928.26 ± 7.29** | **26.88 ± 0.52** | **11.05 ± 0.23** | **15.02 ± 0.35** |





**Table.5** The effective C storage ($C_{eff}$) of Loch Sunart's postglacial and glacial sediment in comparison
to Scottish terrestrial C stores.

| C Inventories | Area (km$^2$) | TC (Mt) | $C_{eff}$ (Mt km$^{-2}$) | $OC_{eff}$ (Mt km$^{-2}$) | $IC_{eff}$ (Mt km$^{-2}$) | Reference |
|---|---|---|---|---|---|---|
| **Postglacial** | | | | | | |
| Inner Basin | 5.5 | 1.31 | 0.238 | 0.191 | 0.047 | |
| Middle Basin | 24.7 | 14.08 | 0.57 | 0.285 | 0.284 | |
| Outer Basin | 17.1 | 4.49 | 0.263 | 0.077 | 0.184 | |
| **Glacial** | | | | | | |
| Inner Basin | 5.5 | 0.81 | 0.147 | 0.044 | 0.104 | |
| Middle Basin | 24.7 | 3.97 | 0.161 | 0.047 | 0.113 | |
| Outer Basin | 17.1 | 2.21 | 0.129 | 0.038 | 0.091 | |
| Postglacial | 47.3 | 19.88 | 0.42 | 0.189 | 0.213 | |
| Glacial | 47.3 | 7.00 | 0.148 | 0.044 | 0.104 | |
| Loch Sunart | 47.3 | 26.88 | 0.568 | 0.234 | 0.318 | |
| **2 m Depth** | | | | | | |
| Peatlands* | 17270 | 1620 | | 0.094 | | Chapman et al., 2009 |
| Organo-Mineral Soil* | | 754 | | | | Bradley et al., 2005 |
| Mineral Soil* | | 498 | | | | |
| **1 m Depth** | | | | | | |
| Peat | 17369 | 813.9 | | 0.047 | | Aitkenhead & Coull ,2016 |
| Alluvial Soil | 1657 | 40.8 | | 0.025 | | |
| Alpine Soil | 3825 | 145.7 | | 0.038 | | |
| Bare Ground | 1672 | 50.5 | | 0.030 | | |
| Brown Earth | 15971 | 590.3 | | 0.037 | | |
| Gley | 15963 | 645.4 | | 0.040 | | |
| Podzol | 18159 | 536.6 | | 0.029 | | |
| Ranker | 2531 | 82.6 | | 0.033 | | |
| Regosol | 437 | 19.0 | | 0.044 | | |


*Both studies calculated the soil C stocks excluding IC data therefore the stocks only represent the OC
held within these stocks.

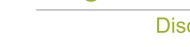
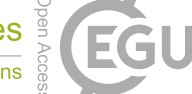


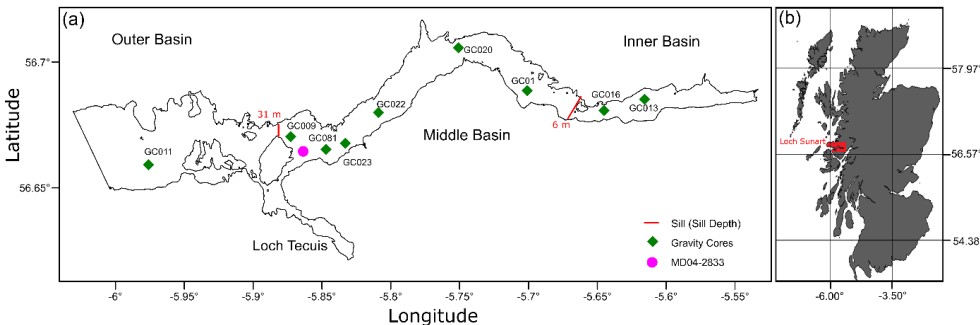

2  **Figure 1**. Maps of Loch Sunart illustrating (a) the three basins and the sediment core

3  locations (b) Loch Sunart in a Scottish context.



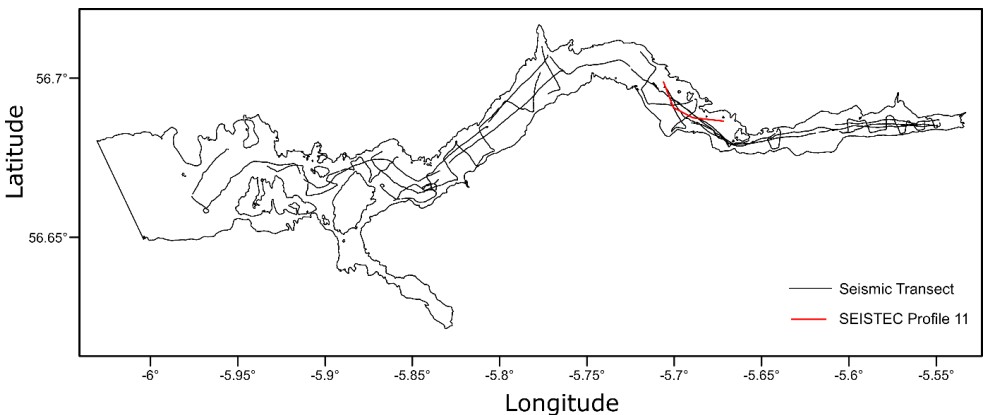

2    **Figure.2.** Map of the 34 Seismic transects undertaken in Loch Sunart with Siestec Profile 11

3    highlighted.




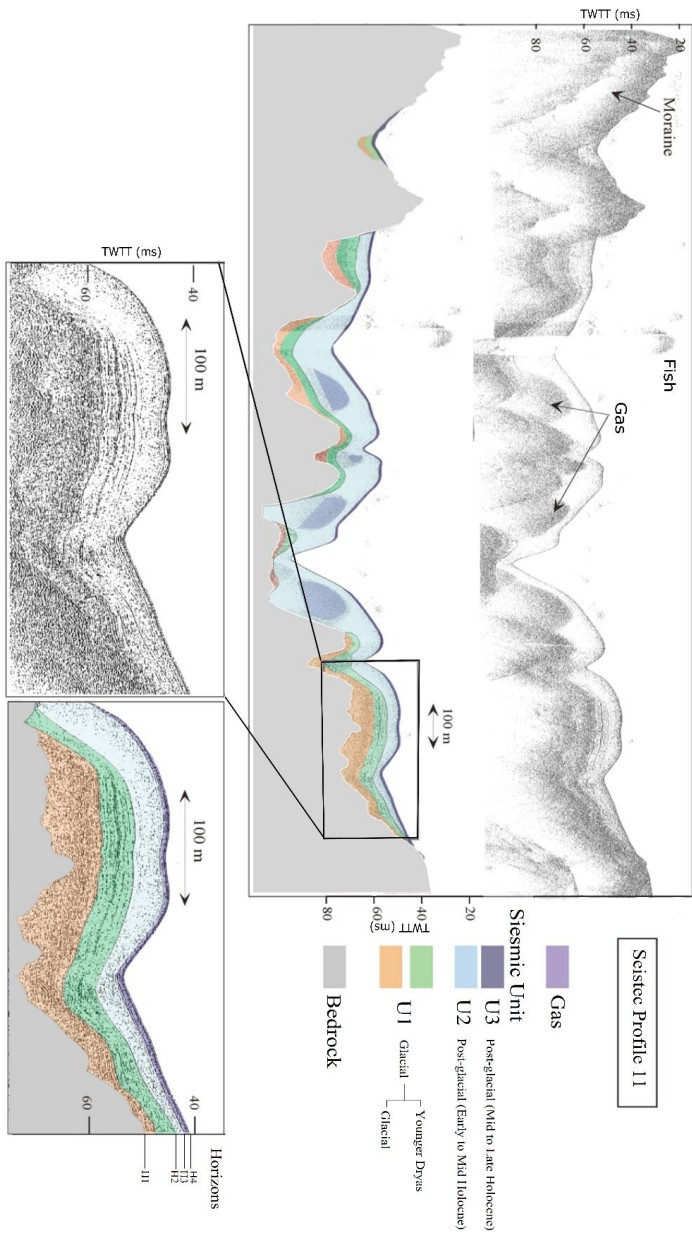

2 **Figure 3.** SIESTEC Profile 11: A characteristic seismic profile displaying the four seismic

3 horizons (H1, H2, H3 and H4) and the three seismic units (U1, U2 and U3) adapted from

4 Baltzer et al.,2010.




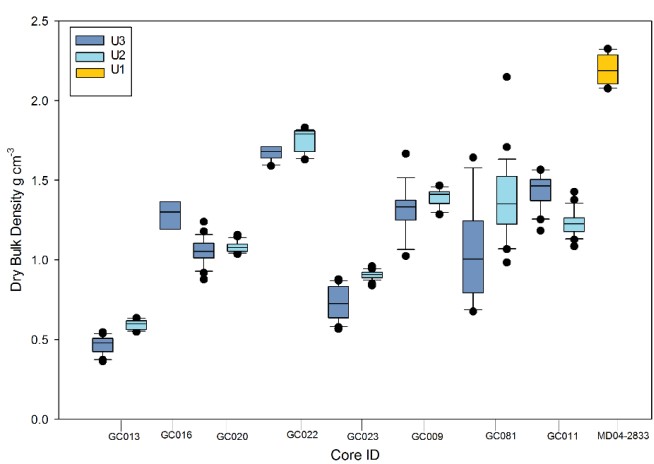

2    **Figure 4.** Dry bulk density values from each sediment cores corresponding to seismic units 1,

3    2 and 3.





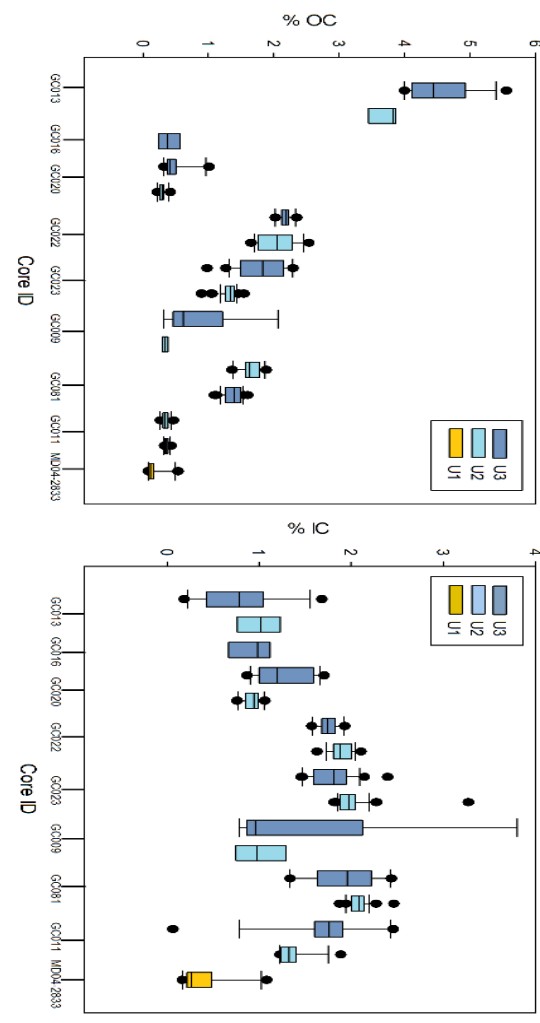

2  **Figure 5.** %OC and %IC values from each sediment cores corresponding to seismic units 1, 2

3  and 3.



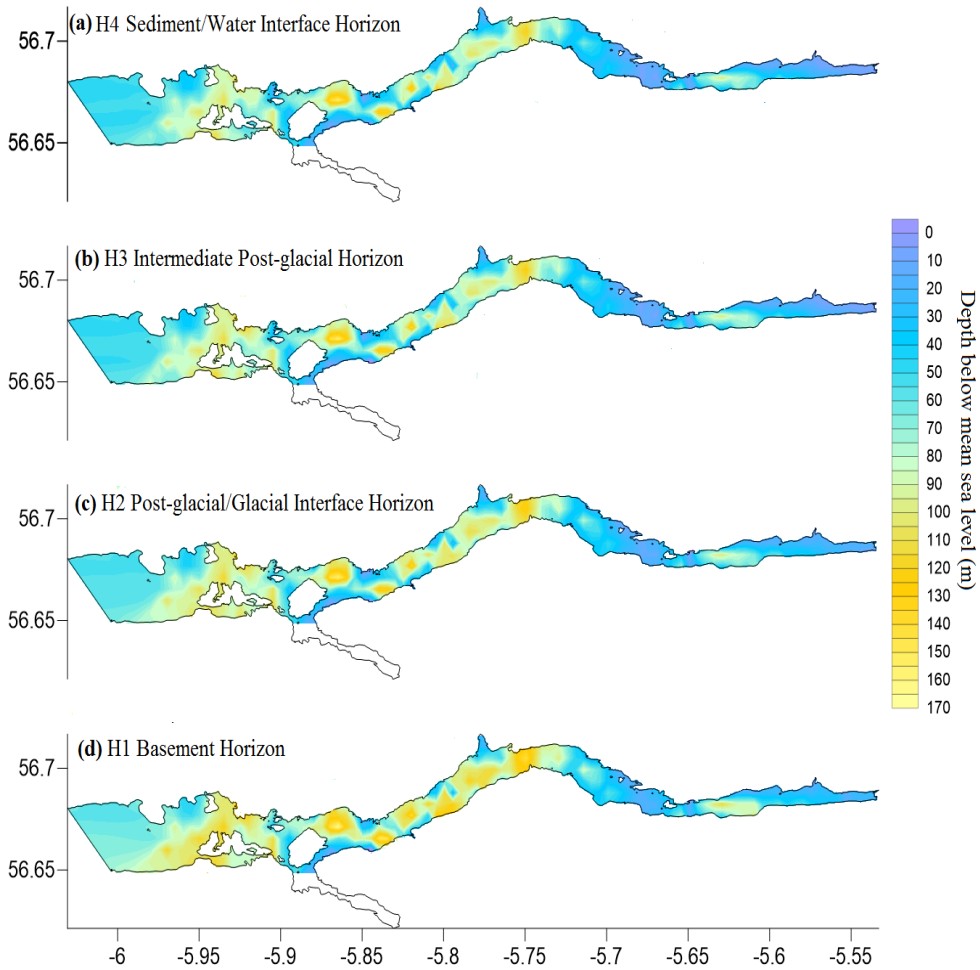

2  **Figure 6.** Contour maps defining the topography of each seismic horizon. **(a)** H4

3  sediment/water interface. **(b)** H3 intermediate post-glacial horizon. **(c)** H2 post-glacial/glacial

4  interface. **(d)** H1 basement.



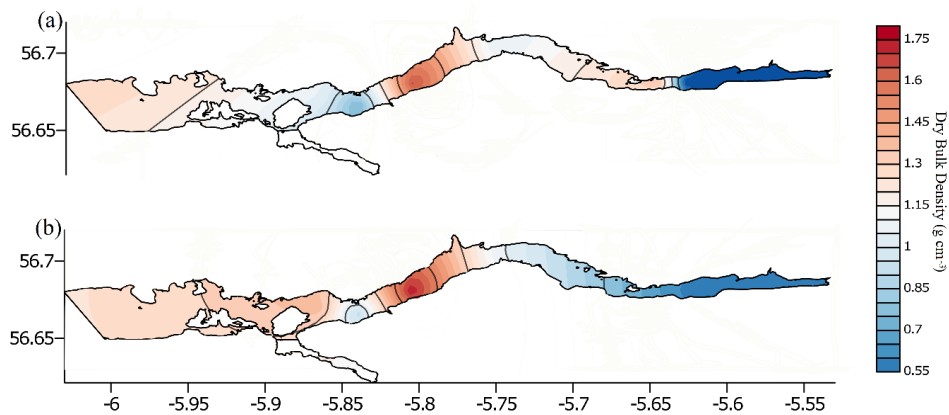

2 **Figure 7.** Contour maps showing the output of the spatial distribution model for the mean

3 dry bulk density of **(a)** U3. **(b)** U2.





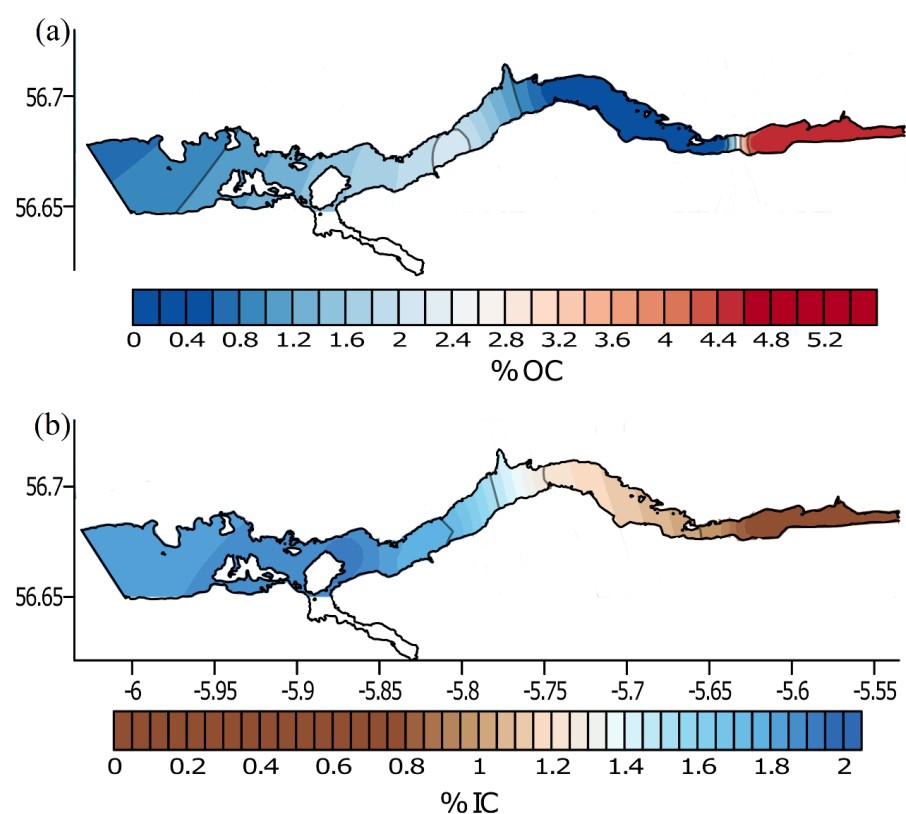

2    **Figure 8.** Output of U3 spatial distribution model for **(a)** Total carbon**. (b)** Organic carbon.

3    **(c)** Inorganic carbon.





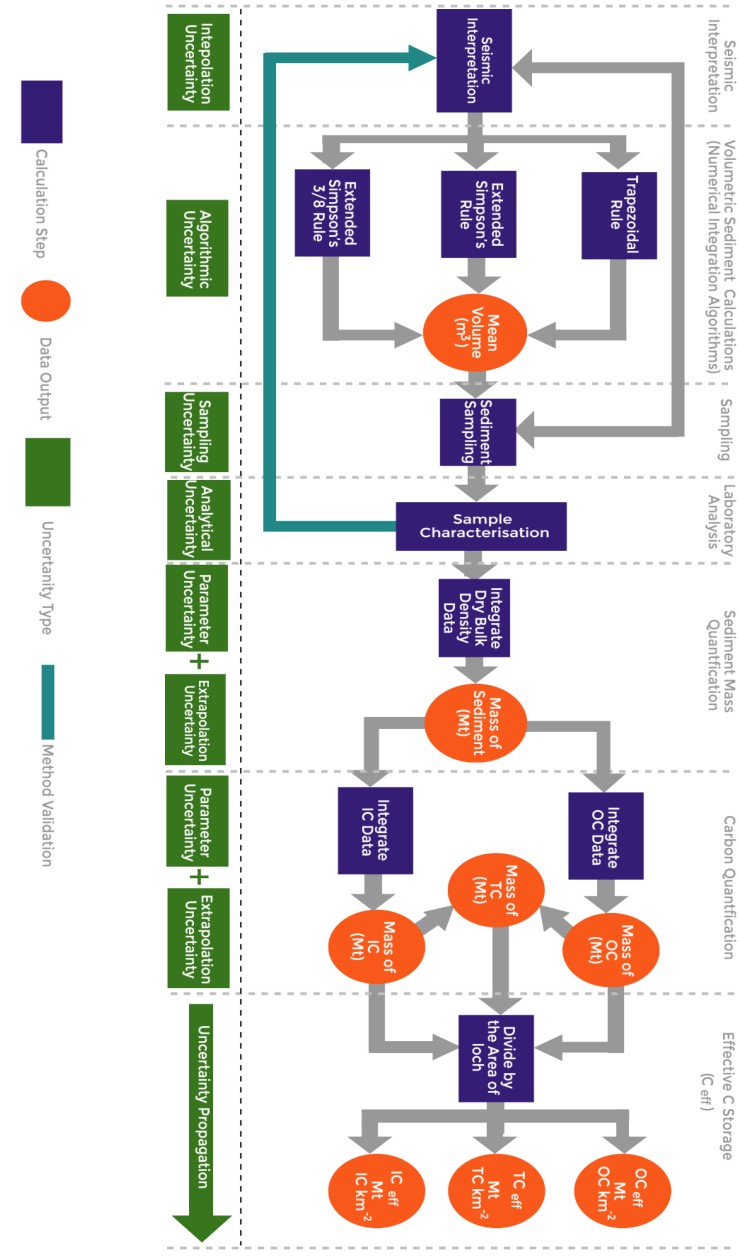

2  **Figure. 9.** Flow diagram detailing the steps towards calculating the sedimentary C stocks

3  within a fjord with the known uncertainties specified.