# Peer review of "Substantial Stores of Sedimentary Carbon held in Mid-Latitude"

_Biogeosciences, 2016_

## Referee Comment (RC1) · Anonymous Referee #1 · 11 Jul 2016

Fjords (lochs) have been shown to be significant global sinks of carbon, especially with respect to their relatively small area on earth. In this study, Smeaton et al. conducted a seismic survey and collected sediment cores from Loch Sunart, a loch system on the west coast of mainland Scotland. The authors stated that lochs on Scotland mainland are comparable to some of the fjords in Norway, Canada, and Fiordland, and therefore, the methods and results from this study could likely be applied in studying other fjord systems. By using seismic data and sediment cores, the authors present a detailed method for calculating the inorganic and organic carbon budget in Loch Sunart. The authors further compared the carbon inventory with the peat & soil inventory in Scotland, which has been well studied. They concluded that Loch Sunart has a much

higher area-normalized burial rate of carbon than peat & soil systems on land, and therefore suggest that lochs should be treated as a standalone system when studying carbon dynamics. The manuscript is well organized and well-written; I believe with some moderate revision, this manuscript would be suitable for publication. Please see the suggested changes below.

Page 1, line 24: in or is? Page 1, line 26: are or area? Page 2, line 5: check citation format Page 3, line 13: What this statement is based on? Page 5, line 18: where or were? Page 6, line 16: Any reference for this method? To what I know, this method is not commonly seen while doing OC analysis. Page 7, line 3: Apparently, SARs are not the same during glacial and interglacial time period, and how do you justify this point? If you use constant SAR from LGM to modern, then you would overestimate Holocene SAR. Page 7, line 14&15: check the sentence. Page 10, section 3.2: the radiocarbon dating of core 2833 in Baltzer et al. (2010) is poor in the lower part of the core. Time dating of 17041 at ∼7 m depth is from LGM, however, there is no evidence showing the dropstone presented at the bottom of the core is whether from H1 or H2. It is likely that core 2833 goes back to ∼25,000 years, especially considering that presence of the fifth horizon in the inner basin & middle basin, which is likely the boundary of LGM (H1). Please justify this point. Page 11, line 18: What is this? Page 12, line 12: The number of decimals might be too high, unless if you can justify it. Page 12, line 17: check gramma Page 13, line 20: How the negative number is calculated? Page 13, line 20&21: This sentence is not supported by the data. Page 14, line 25&26: and also area < 10 m deep? Discussion: Although this paper is more of a methodology paper, I still think it might be interesting to make comparison to some other fjords globally in the aspects of sedimentation rates, carbon accumulation rates, and distribution pattern of carbon along fjords. Smith et al. (2015) summarized all global fjords, however, there is no OC accumulation rates from Scottish fjords yet. By making a comparison of the results in this manuscript to other fjords, even just fjords in NW Europe, it would make the paper more interesting. I would assume Scandinavian fjords would be different to Scottish fjords in carbon accumulation rates, etc. Page 15, line 23: check gramma

Page 16, line 25: What defines long-term here? Based on the data, the loch sediment chronology only goes back to ∼17,000 yrs. Fig. 6: not sure if it is better to show the thickness of each unit, other than the depth of each horizon because the thickness is more interesting and more related while calculating carbon budgets. Fig. 8 is not cited in the manuscript.

—————————————————

---

## Referee Comment (RC2) · J. Hinojosa (Referee) · 21 Jul 2016

This study by Smeaton et al. presents an interesting new methodology to calculate carbon stocks in coastal environments, especially fjords (AKA lochs). By integrating geophysical data with geochemical constraints, the authors estimate amounts of organic and inorganic carbon stored in Loch Sunart in Scotland. Interestingly, their estimates suggest orders of magnitude more carbon stored in Scottish lochs than previous estimates would suggest.

The most important contribution of this work is the authors' effort to make the methods reproducible. They urge scientists to calculate carbon stocks in similar environments using similar datasets, which would greatly improve our understanding and quantification of carbon stored in coastal environments. For that reason, as well as for the solid scientific method and good writing/organization of the study, I believe this manuscript should be published in the relevant journal of Biogeosciences after addressing some moderate comments below.

I have two broader critiques of the paper, as well as a few minor comments, which are addressed by line number below. First, while this paper does a fantastic job of calculating carbon stocks in Loch Sunart, I don't believe the authors give enough credit to other work that has preceded this paper. While this may indeed be the first time that the total carbon stock in a fjord has been estimated, as the authors claim, there have been many studies that calculate carbon fluxes in similar environments. For example, there is no reference to Sepulveda et al. (2011), which did a comprehensive assessment of organic carbon flux in Chilean fjords based on surface sediments, or Walsh et al. (1991), which used sediment trap data to calculate carbon burial rates on continental margins. Furthermore, though the authors cite Hinojosa et al. (2014), they do not acknowledge the work therein that generates carbon accumulation rates based on a suite of sediment cores and calculates regional C fluxes for the entire fjord region. Smeaton et al. acknowledge the Scottish carbon cycling literature very well, but I encourage the authors to do a bit more digging in similar fjordic systems around the world.

Second, I'm not sure I agree that the estimates of carbon stocks can be presented with such precision. I greatly approve of the authors' effort to quantify uncertainty, but even they acknowledge that there are other, unquantifiable sources of uncertainty (page 14, lines 15-16). I believe that there are even more sources of unknown uncertainty, especially with the extrapolation of geochemical data to large volumes of sediment. Unfortunately, there isn't a way to constrain the unknown unknowns. For this reason, I would feel far more comfortable if fewer significant figures were used in all the various estimates. In particular, the final carbon stock estimates of $26.88 \pm 0.52$ Mt C, split between $11.05 \pm 0.23$ Mt OC and $15.02 \pm 0.35$ IC, should be presented as something more along the lines of $27 \pm 0.5$, $11 \pm 0.2$, and $15 \pm 0.4$. I don't believe this methodol-

ogy can provide the precision of the initially reported carbon stocks. However, it doesn't detract at all from the importance of the paper to provide broader estimates that reflect the extra uncertainties that cannot be quantified. Overall, this is a great piece of work and another quantitative study that shows how important – and underappreciated – fjords/lochs are in the context of global carbon cycling and storage.

Sincerely, Jessica Hinojosa

Minor comments:

Page 1, Line 26: "are" should be "area" Page 3, Line 6: See first comment above...I think the effort to quantify C fluxes deserves recognition to this end Page 3, Line 12: Why would only using the top 10 cm lead to an underestimation, specifically? Page 4, Line 21: Fiordland shouldn't have "the" in front of it Page 5, Line 19: "where" should be "were" Page 7, Line 14: same comment as above Page 8, formulas; What does Ai represent? Page 10, Lines 18-19: You mention that this core has dating problems, but you use it anyway in all calculations. Can you speculate on why there are these discrepancies, or why you justified using the data regardless? Page 10, Lines 24-27: As per my second major comment above, I don't think two-point linear interpolations can be reported with this amount of precision; there is obviously going to be much more significant deviation from this number over time. Page 11, Line 18: Delete this line Page 12, Section 3.3.4: Use scientific notation rather than reporting long numbers Page 14, Lines 5-8: Don't totally agree that a few numbers per core means the mean $\pm$ SD is representative of the entire sediment or seismic unit. Page 14, Line 17: "socks" should be "stocks" Page 15, line 2: Not probably, definitely! See references above, plus Smith et al. (2015) Page 16, Lines 24-25: Not true...most fjord basins get scoured by glacial advances during ice ages, which dumps interglacial sediment into the offshore environment. Figure 5: Images seem stretched, but that may be an artifact of the manuscript submission. Just double-check before final draft. Figures 7 and 8: It would be nice to see the locations where there are data (from Figure 1) overlain on these models.

Sepúlveda, J., Pantoja, S., Hughen, K.A., 2011. Sources and distribution of organic matter in northern Patagonia fjords, Chile (âĹij44–47°S) A multi-tracer approach for carbon cycling assessment. CSR 31, 315–329. doi:10.1016/j.csr.2010.05.013 Walsh, J.J., 1991. Importance of continental margins in the marine biogeochemical cycling of carbon and nitrogen.
* * *

---

## Author Comment (AC1) · 24 Aug 2016

Referee Comment 1 Fjords (lochs) have been shown to be significant global sinks of carbon, especially with respect to their relatively small area on earth. In this study, Smeaton et al. conducted a seismic survey and collected sediment cores from Loch Sunart, a loch system on the west coast of mainland Scotland. The authors stated that lochs on Scotland mainland are comparable to some of the fjords in Norway, Canada, and Fiordland, and therefore, the methods and results from this study could likely be applied in studying other fjord systems. By using seismic data and sediment cores, the authors present a detailed method for calculating the inorganic and organic carbon budget in Loch Sunart. The authors further compared the carbon inventory with the

peat & soil inventory in Scotland, which has been well studied. They concluded that Loch Sunart has a much higher area-normalized burial rate of carbon than peat & soil systems on land, and therefore suggest that lochs should be treated as a standalone system when studying carbon dynamics. The manuscript is well organized and well-written; I believe with some moderate revision, this manuscript would be suitable for publication. Please see the suggested changes below.

*We thank the referee for these very positive comments; it is encouraging to note the external peer-review support for our arguments to suggest the stand-alone nature of these global systems and their underestimated significance as carbon sinks in comparison to adjacent terrestrial sinks.

Page 1, line 24: in or is?

*is

Page 1, line 26: are or area?

*area

Page 2, line 5: check citation format

*& changed to and

Page 3, line 13: What this statement is based on?

*Reference to a paper by Norgaard-Pedersen et al. (2005) has been added to highlight the fact that data exist to show significant amounts of IC present in these buried sediment sequences.

Page 5, line 18: where or were?

*were

Page 6, line 16: Any reference for this method? To what I know, this method is not commonly seen while doing OC analysis.

*Reference to a paper by Loh et al. 2008 has been added, outlining the methodology used.

Page 7, line 3: Apparently, SARs are not the same during glacial and interglacial time period, and how do you justify this point? If you use constant SAR from LGM to modern, then you would overestimate Holocene SAR.

*This is true, but one of the reasons that we did not focus on SARs, was that the constraining chronologies from the available sediment cores did not provide good constraints on the glacial and deglacial portions of the stratigraphy and therefore one must acknowledge that the uncertainties will be large. However, if one adopts an approach based on well-defined seismic units, as we have done, there is far less ambiguity in the methodology.

Page 7, line 14&15: check the sentence.

*were.

Page 10, section 3.2: the radiocarbon dating of core 2833 in Baltzer et al. (2010) is poor in the lower part of the core. Time dating of 17041 at âĹij7 m depth is from LGM, however, there is no evidence showing the dropstone presented at the bottom of the core is whether from H1 or H2. It is likely that core 2833 goes back to âĹij25,000 years, especially considering that presence of the fifth horizon in the inner basin & middle basin, which is likely the boundary of LGM (H1). Please justify this point.

*We thank the referee for these interesting suggestions, but would note that current understanding of the regional ice sheet dynamics for this sector of the NW European shelf would support the refree's inferences. First, the reported age from MD04-2833 is a calibrated age of 17041 +/-312 cal BP; this is consistent with the known timing of regional ice stream deglaciation back towards the present coastline (Clark et al., 2012). The reported age is therefore consistent with the transition from ice grounding within the fjord to a marine, ice-free environment. The referee speculates whether or

not "clasts" within the underlying diamict below this basal marine age of 17041 +/-312 cal BP may represent "dropstones" associated with Heinrich Layers. Given our current understanding of the last British and Irish Ice Sheet dynamics (e.g. Scourse et al., 2009), the possibility of Heinrich or Heinrich-like events at this time within these fjord environments is extremely unlikely, since they represent the major conduits for the ice streams which extended as far offshore as the continental shelf break.

Page 11, line 18: What is this?

*Line deleted as it was from previous version of paper.

Page 12, line 12: The number of decimals might be too high, unless if you can justify it.

*The figures have been changed to scientific notation rounded to 2 decimal places.

Page 12, line 17: check gramma

*We have corrected the grammatical construction of this sentence.

Page 13, line 20: How the negative number is calculated?

*This was a typo, which has now been removed; this is not a negative number.

Page 13, line 20&21: This sentence is not supported by the data.

*We have corrected these numbers, which should read as follows in terms of their OC:IC ratios 4, 1 and 0.42.

Page 14, line 25&26: and also area < 10 m deep?

*We don't understand the referee's comment; we refer to a published study (Burrows et al., 2014) where the authors only accounted for sedimentary C stock within the surface 10 cm – our point is that this approach grossly underestimates the full depth sediment C stock.

Discussion: Although this paper is more of a methodology paper, I still think it might

be interesting to make comparison to some other fjords globally in the aspects of sedimentation rates, carbon accumulation rates, and distribution pattern of carbon along fjords. Smith et al. (2015) summarized all global fjords, however, there is no OC accumulation rates from Scottish fjords yet. By making a comparison of the results in this manuscript to other fjords, even just fjords in NW Europe, it would make the paper more interesting. I would assume Scandinavian fjords would be different to Scottish fjords in carbon accumulation rates, etc.

*We thank the referee for this very constructive comment; we had originally intended the focus of this manuscript to be methodological, but we are persuaded that the global comparisons to which the referee alludes would be usefully incorporated. Therefore, we have revised the manuscript accordingly (please see similar suggestions from referee 2, where we provide the detailed information on our responses)

Page 15, line 23: check grammar

*Brackets and question mark removed as they were part of a previous version of the paper.

Page 16, line 25: What defines long-term here? Based on the data, the loch sediment chronology only goes back to âĹij17,000 yrs.

*To avoid potential confusion, we have removed this statement and instead added text to describe glacial/interglacial cycling processes on erosion/deposition to the adjacent shelf/slope.

Fig. 6: not sure if it is better to show the thickness of each unit, other than the depth of each horizon because the thickness is more interesting and more related while calculating carbon budgets.

*We acknowledge that the underlying data in figure 6 represent an important step in our methodology to calculate sediment volumes; unfortunately – the programme used to generate these plots does not easily lend itself to generating the thickness plots which

the referee mentions. For this reason, we agree with the referee that the differences in unit thicknesses are not immediately apparent and have therefore moved figure 6 to the supplementary materials section (where it can be viewed to understand the underlying methodology).

Fig. 8 is not cited in the manuscript.

*This figure is now cited in the text on page 12 line 24.

Please also note the supplement to this comment:
http://www.biogeosciences-discuss.net/bg-2016-245/bg-2016-245-AC1-supplement.pdf

---

## Author Comment (AC2) · 24 Aug 2016

Referee 2 – J. Hinojosa

This study by Smeaton et al. presents an interesting new methodology to calculate carbon stocks in coastal environments, especially fjords (AKA lochs). By integrating geophysical data with geochemical constraints, the authors estimate amounts of organic and inorganic carbon stored in Loch Sunart in Scotland. Interestingly, their estimates suggest orders of magnitude more carbon stored in Scottish lochs than previous estimates would suggest. The most important contribution of this work is the authors' effort to make the methods reproducible. They urge scientists to calculate carbon stocks in similar environments using similar datasets, which would greatly improve our understanding and quantification of carbon stored in coastal environments. For that reason, as well as for the solid scientific method and good writing/organization of the study, I believe this manuscript should be published in the relevant journal of Biogeosciences after addressing some moderate comments below.

*We thank the reviewer for the very helpful review, which highlights the significance of the revised stock estimates and rigorous methodology adopted.

I have two broader critiques of the paper, as well as a few minor comments, which are addressed by line number below. First, while this paper does a fantastic job of calculating carbon stocks in Loch Sunart, I don't believe the authors give enough credit to other work that has preceded this paper. While this may indeed be the first time that the total carbon stock in a fjord has been estimated, as the authors claim, there have been many studies that calculate carbon fluxes in similar environments. For example, there is no reference to Sepulveda et al. (2011), which did a comprehensive assessment of organic carbon flux in Chilean fjords based on surface sediments, or Walsh et al. (1991), which used sediment trap data to calculate carbon burial rates on continental margins. Furthermore, though the authors cite Hinojosa et al. (2014), they do not acknowledge the work therein that generates carbon accumulation rates based on a suite of sediment cores and calculates regional C fluxes for the entire fjord region. Smeaton et al. acknowledge the Scottish carbon cycling literature very well, but I encourage the authors to do a bit more digging in similar fjordic systems around the world.

*First, we have revised the emphasis of the manuscript to highlight the wider, global significance of the work in estimating coastal carbon sediment stocks. We have also included reference to the papers mentioned (and some others), which we believe will make the manuscript a very useful reference (as well as methodological study). In particular, we have prepared and include a new summary table (Table 6) which shows the accumulation and burial rates from a wide range of fjords – this allows a new discussion of the Loch Sunart mid-latitude data in comparison to other fjord settings, including the high latitude systems.

Second, I'm not sure I agree that the estimates of carbon stocks can be presented with such precision. I greatly approve of the authors' effort to quantify uncertainty, but even they acknowledge that there are other, unquantifiable sources of uncertainty (page 14, lines 15-16). I believe that there are even more sources of unknown uncertainty, especially with the extrapolation of geochemical data to large volumes of sediment. Unfortunately, there isn't a way to constrain the unknown unknowns. For this reason, I would feel far more comfortable if fewer significant figures were used in all the various estimates. In particular, the final carbon stock estimates of 26.88 $\pm$ 0.52 Mt C, split between 11.05 $\pm$ 0.23 Mt OC and 15.02 $\pm$ 0.35 IC, should be presented as something more along the lines of 27 $\pm$ 0.5, 11 $\pm$ 0.2, and 15 $\pm$ 0.4. I don't believe this methodology can provide the precision of the initially reported carbon stocks. However, it doesn't detract at all from the importance of the paper to provide broader estimates that reflect the extra uncertainties that cannot be quantified. Overall, this is a great piece of work and another quantitative study that shows how important – and underappreciated – fjords/lochs are in the context of global carbon cycling and storage.

*We agree with the referee and have adjusted the precision with which we report these data. The nature of our study inevitably gives rise to "unknowns", we feel that the discussion and Figure 8, in particular, go a long way to addressing some of these "unknowns" and that this sets a very clear outline of our current understanding.

Page 1, Line 26: "are" should be "area"

*Area.

Page 3. Line 6: See first comment above. . .I think the effort to quantify C fluxes deserves recognition to this end.

*Sentences added recognising the efforts to quantify C fluxes in global fjords (NZ, Canada, Chile, Alaska and NW Europe.

Page 3, Line 12: Why would only using the top 10 cm lead to an underestimation,

specifically?

*We have clarified the sentence to explain that a study of only the surficial 10 cm will, in systems with sometimes >25 m of sediment, lead to significant underestimation of the stock.

Page 4, Line 21: Fiordland shouldn't have "the" in front of it. *"the" removed.

Page 5, Line 19: "where" should be "were" .

*Changed to were.

Page 7, Line 14: same comment as above .

*Changed to were.

Page 8, formulas; What does Ai represent?

*Ai represents the abscissa which is the distance from a point to the vertical or y -axis, measured parallel to the horizontal or x -axis; the x –coordinate. This definition has been added to the text.

Page 10, Lines 18-19: You mention that this core has dating problems, but you use it anyway in all calculations. Can you speculate on why there are these discrepancies, or why you justified using the data regardless?

*The calculations used no longer include this core; we have added a note to this effect in the text. We feel it is still useful to highlight that there are sometimes dating problems (or discrepancies) with marine sediment cores and are sufficiently confident in our remaining data to highlight this fact.

Page 10, Lines 24-27: As per my second major comment above, I don't think two-point linear interpolations can be reported with this amount of precision; there is obviously going to be much more significant deviation from this number over time. *We have changed the text.

Page 11, Line 18: Delete this line.

*Line deleted.

Page 12, Section 3.3.4: Use scientific notation rather than reporting long numbers.

*The figures have been changed to scientific notation rounded to 2 decimal places.

Page 14, Lines 5-8: Don't totally agree that a few numbers per core means the mean $\pm$ SD is representative of the entire sediment or seismic unit.

*We agree and have removed reference to "entire" in this sentence. Our coring programme and sampling represent our best effort to acknowledge this issue i.e. we have sampled at different depths within a stratigraphic sequence and at different locations within the fjord system. Within the limitations of sampling, we argue that these data are representative. While the mean and SD values are quoted for these units, we do also include the range of values in, for example, Figures 4 & 5 (and a reader can see the minima and maxima in the supplementary materials section). Our entire methodology then relies on propagating the "uncertainties" into the final stock estimates.

Page 14, Line 17: "socks" should be "stocks" .

*Socks changed to stocks!

Page 15, line 2: Not probably, definitely! See references above, plus Smith et al. (2015).

*Removed probably from the sentence.

Page 16, Lines 24-25: Not true. . .most fjord basins get scoured by glacial advances during ice ages, which dumps interglacial sediment into the offshore environment.

*We agree that over interglacial/glacial timescales these C stores are vulnerable to scouring through the advance of glaciers. The original sentence described vulnerability during the interglacial period. To avoid potential confusion, we have removed this

statement and instead added text to describe glacial/interglacial cycling processes on erosion/deposition to the adjacent shelf/slope.

Figure 5: Images seem stretched, but that may be an artefact of the manuscript submission. Just double-check before final draft.

*Some of the diagrams may appear stretched - this is indeed an artefact from importing them into a word document. This should not be problematic in any final version.

Figures 7 and 8: It would be nice to see the locations where there are data (from Figure 1) overlain on these models.

*We agree that it would be a useful point of reference to have the position of the sediment cores overlain on Figures. 7 and 8. The figures have been altered to reflect this suggestion and these are now included.

Please also note the supplement to this comment:
http://www.biogeosciences-discuss.net/bg-2016-245/bg-2016-245-AC2-supplement.pdf

---

## Author Response (AR1)

Dear Dr Pantoja

Firstly thank you for talking the time to handle this manuscript submission.

We have revised the paper in accordance with both reviewers' comments. Both reviewers commented that the paper could be more outward looking and should include carbon accumulation and burial rates therefore a new section has been added which details these rates for Loch Sunart. Through this alteration we have been able to compare Loch Sunart to fjords globally and for the first time cautiously estimate sedimentary IC accumulation in a fjord, we believe these changes have improved the manuscript and increased its appeal.

We agree with your comment that the title is too technical and does not convey the importance of the methodology or the results. Therefore we have changed the title to "Substantial Stores of Sedimentary Carbon held in Mid-Latitude Fjords" we believe this is a more impactful title which will appeal to a wider readership than the original.

Craig Smeaton
(On Behalf of the Authors)

[revised manuscript text omitted]
⁻¹) Min | Max | OC Burial Rate (g m⁻² y⁻¹) Min | Max | OC Burial (Tonnes yr⁻¹) Min | Max | Reference |
|---|---|---|---|---|---|---|---|---|---|
| Loch Sunart | 47.3 | 0.017-0.089 | 3.0 | 32.1 | 1.89[a] | 25.68[b] | 8.9 x 10¹ | 1.2 x 10³ | This Study |
| *NW Europe/Artic* | | | | | | | | | |
| Loch Creran | 13.3 | 0.2-0.5 | | | 21.9 | 193.45 | 2.9 x 10² | 2.6x 10³ | Loh et al., 2008 |
| Nordasvannet Fjord | 4.6 | | | | | 2.2 | | 1.0 x 10¹ | Winkelmann and Knies 2005 |
| Storfjord | 1424 | | | | 21.0 | 40.0 | 3.0 x 10⁵ | 5.7 x 10⁵ | Müller. 2001 |
| | 9 | | | | | | | | |
| Kongsfjorden | 817 | | | | 9 | 13 | 7.4 x 10³ | 1.0 x 10⁴ | Kulinski et al., 2014 |
| *Canada/Alaska* | | | | | | | | | |
| Saguenay Fjord | 360 | | | | 24.5 | 291.0 | 8.8 x 10³ | 1.0 x 10⁵ | St-Onge and Hillaire-Marcel. 2001 |
| Vegetated Alaskan Fjords | | | | | 13 | 82 | | | Cui et al., 2016 |
| Glaciated Alaskan Fjords | | | | | 30 | 1113 | 5.7 x 10⁵ | 7.6 x 10⁵ | |
| *Chile* | | | | | | | | | |
| Jacaf Fjord | 236 | 0.28 | 33.4 | 40.8 | 21.0 | 25.7 | 5.0 x 10³ | 6.1 x 10³ | Sepúlveda et al., 2011 |
| Ventisquero Sound | 7.2 | 0.74 | 69.3 | 82.5 | 43.7 | 52.0 | 3.1 x 10² | 3.7 x 10² | |
| Puyuhuapi Fjord | 111 | 0.25 | 11.0 | 34.2 | 6.9 | 21.6 | 3.1 x 10³ | 9.6 x 10³ | |
| Aysen Fjord | 340 | 0.24 | 10.5 | 20.7 | 6.6 | 13.1 | 2.3 x 10³ | 4.4 x 10³ | |
| Quitralco Fjord | 116 | 0.47 | 4.6 | 55.3 | 2.9 | 34.8 | 3.3 x 10² | 4.0 x 10³ | |
| Cupquelan Fjord | 125 | 0.14 | 1.9 | 8.4 | 1.2 | 5.3 | 1.5 x 10² | 6.6 x 10² | |
| *New Zealand* | | | | | | | | | |
| Milford Sound | 25.3 | 0.268 | 23.2 | | 18.6 | | 4.7 x 10² | | Knudson et al., 2011 |
| George Sound | 32.9 | 0.087 | 3.63 | | 2.90 | | 9.5 x 10¹ | | |
| Thompson Sound | 49.3 | 0.113 | 10.6 | | 8.48 | | 4.18 x 10² | | |
| Nancy Sound | 13.9 | 0.204 | 32.6 | | 26.1 | | 3.62 x 10² | | Pickrill. 1993 |
| Doubtful Sound | 83.7 | 0.079 | 23.2 | | 18.6 | | 1.6 x 10³ | | |
| Breaksea Sound | 61.5 | 0.038 | 9.07 | | 7.26 | | 4.5 x 10² | | Smith et al., 2015 |
| Dusky Sound | 181 | 0.012 | 2.31 | | 1.85 | | 3.3 x 10² | | |
| Long Sound | 93 | 0.094 | 16.0 | | 12.8 | | 1.2 x 10³ | | |
| Dusky Sound | 181 | 0.16 | 44 | 68 | 35.2[b] | 54.4[b] | 6.4 x 10³ | 9.8 x 10³ | Hinojosa et al., 2014 |
| Doubtful Sound | 83.7 | 0.38 | 115 | 169 | 92[b] | 135.2[b] | 7.7 x 10³ | 1.1 x 10⁴ | |
| George Sound | 32.9 | 0.10 | 4.8 | | 3.84[b] | | 1.3 x 10² | | |
| Thompson Sound | 49.3 | 0.06-0.17 | 15.2 | | 12.16[b] | | 6.0 x 10² | | |

[a]OC Burial rate calculated assuming a burial efficiency of 63% (Sepúlveda et al., 2005).

[b]OC Burial rate calculated assuming a burial efficiency of 80% (Smith et al., 2015).

[Figure]

**Figure. -1**. Maps of Loch Sunart illustrating (a) the three basins and the sediment core locations (b) Loch Sunart in a Scottish context.

[Figure]

**Figure. 2.** Map of the 34 Seismic transects undertaken in Loch Sunart with Siestec Profile 11

highlighted.

[Figure]

**Figure. 3.** SIESTEC Profile 11: A characteristic seismic profile displaying the four seismic horizons (H1, H2, H3 and H4) and the three seismic units (U1, U2 and U3) adapted from Baltzer et al.,2010.

[Figure]

**Figure. 4.** Dry bulk density values from each sediment cores corresponding to seismic units 1,

2 and 3.

[Figure]

**Figure. 5.** %OC and %IC values from each sediment cores corresponding to seismic units 1, and 3.

[Figure]

**Figure. 6.** Contour maps showing the output of the spatial distribution model for the mean dry bulk density of **(a)** U3. **(b)** U2. Sampling locations indicated with black diamonds.

[Figure]

**Figure. 7.** Output of U3 spatial distribution model for **(a)** Organic carbon. **(b)** Inorganic carbon.

Sampling locations indicated with black diamonds.

[Figure]

**Figure. 8.** Flow diagram detailing the steps towards calculating the sedimentary C stocks within a fjord with the known uncertainties specified.